

# Composition and variability of gaseous organic pollution in the port megacity of Istanbul: source attribution, emission ratios and inventory evaluation

Baye T.P. Thera[1], Pamela Dominutti[2], Fatma Öztürk[3], Thérèse Salameh[4], Stéphane Sauvage[4], Charbel Afif [5,6], Banu Çetin[6], Cécile Gaimoz[7], Melek Keleş[2], Stéphanie Evan[8], Agnès Borbon[1]

[1]Laboratoire de Météorologie Physique, CNRS-Université Clermont Auvergne, UMR6016, 63117, Clermont Ferrand, France.
[2]Wolfson Atmospheric Chemistry Laboratories, Department of Chemistry, University of York, Heslington, York, YO10 5DD, UK.
[3]Environmental Engineering Department, Bolu Abant Izzet Baysal University (BAIBU), 14030, Bolu, Turkey.
[4]IMT Lille Douai, Univ. Lille, SAGE - Département Sciences de l'Atmosphère & Génie de l'Environnement, 59000 Lille, France.
[5]Emissions, Measurements, and Modeling of the Atmosphere (EMMA) Laboratory, GEP Unit, Centre For Analysis and Research, Faculty of Sciences, Saint Joseph University, Beirut, Lebanon.
[6]The Cyprus Institute, EEWRC, Nicosia, Cyprus
[7]Laboratoire Interuniversitaire des Systèmes Atmosphériques (LISA) Créteil, France.
[8]LACY, Ile de La Réunion.

*Correspondence to*: Baye T.P. Thera (baye_toulaye_pehan.thera@uca.fr) and Agnès Borbon (Agnes.BORBON@uca.fr).

**Abstract**

In the framework of the TRANSport Emissions and Mitigation in the East Mediterranean (TRANSEMED/ChArMEx) program; Volatile Organic Compound (VOC) measurements were performed for the first time in Istanbul (Turkey) at an urban site in September 2014. One commercial gas-chromatograph coupled to a flame ionization detector (GC-FID) and one proton transfer mass spectrometer (PTR-MS) were deployed. In addition, sorbent tubes and canisters were implemented within the megacity close to major emission sources. More than 70 species including non-methane hydrocarbons (NMHC), oxygenated 25  VOCs (OVOC) and organic compounds of intermediate volatility (IVOC) have been quantified. Among these compounds, 23 anthropogenic and biogenic species were continuously collected at the urban site.

VOC concentrations show a great variability with maxima exceeding 10 ppb (i.e. n-butane, toluene, methanol, and acetaldehyde) and mean values between 0.1 (methacrolein+methylvinylketone) to 4.9 ppb (methanol). OVOC represents 43.9 % of the total VOC concentrations followed by alkanes (26.3 %), aromatic compounds (20.7 %), alkenes (4.8 %), terpenes 30  (3.4 %) and acetonitrile (0.8 %). However, on average, the atmospheric composition of anthropogenic alkanes and aromatics is similar to the ones of European megacities like Paris and London, suggesting the impact of traffic emissions for those compounds. Unusual diurnal profiles of anthropogenic VOC, different from the one of traffic derived products like NOx, reveal the complex interaction between emissions and meteorology. Multiple evidences of the impact of sources other than traffic like industrial activities under continental and south-southwesterly wind regimes or ship emissions on IVOC loads was 35  found.



Five factors have been extracted from the PMF model (EPA/PMF 5.0) and have been compared to source profiles established by near-field measurements and other external variables (meteorological parameters, NOx, CO, $SO_2$…). Surprisingly, road transport is not the dominant source by only explaining 15.8 % of measured VOC concentrations contrary to the local emission inventory. Other factors are toluene from solvent use (14.2 %), biogenic terpenes (7.8 %), natural gas evaporation (25.9 %), composed of butanes, and a last factor characterized by mixed regional emissions and composed of most of the species (36.3 %). The PMF results point out the influence of industrial emissions while there is no clear evidence of the impact of ship emissions on the measured VOC distribution. One reason might be the absence of IVOC in the input matrix; these compounds should be considered in future studies dealing with coastal urban areas. The sensitivity of PMF results on input data (time resolution, meteorological period, peak episode, uncertainty) was tested. While some PMF run are statistically less performant than the reference run, sensitivity tests show that same factors (number and type) are found with slightly different factor contributions (up to 15 % of change).

Finally, the emission ratios (ER) of VOC relative to carbon monoxide (CO) were established. These ratios are consistent with those observed in Los Angeles within a factor of 2. These ER and the road transport factor from PMF were used to estimate VOC emissions and to evaluate three downscaled global emissions inventories (EDGAR, ACCMIP and MACCity). It was found that OVOC emissions were underestimated by a factor of 10 to 26 depending on the inventory. NMHC emission estimations were most of the time within the same range or overestimated by a 3 to 26-fold. In the road transport emission evaluation, EDGAR inventory was found to be better than ACCMIP inventory with most of the compounds within a range of 3 except for

Like in TRANSEMED-Beirut (Lebanon), our work stresses the inadequacy of global emission inventories in the East Mediterranean and discrepancies between emission inventories themselves. There is an urgent need to better represent VOC emissions in this region including non-traffic sources, OVOC and lower volatility organic compounds. VOC emissions are expected to be much larger than expected and larger than Europe and North America.

## 1 Introduction

Clean air is a vital need for all living beings. However, air pollution continues to pose a significant threat to health worldwide (WHO, 2005; Nel, 2005) air quality, climate change, ecosystems (crop yield loss, acidity of ecosystem (Matson et al., 2002), and buildings corrosion (Primerano et al., 2000). The World Health Organization (WHO) estimates that 4.2 million people die every year as a result of exposure to ambient (outdoor) air pollution and 3.8 million people from exposure to smoke from dirty cook stoves and fuels, and 91 % of the world's population lives in places where air quality exceeds WHO guidelines limits (WHO,2018).

Among the various types of air pollutants in the atmosphere, Volatile Organic Compounds (VOC) includes hundreds of species grouped in different families (alkanes, alkenes, aromatics, alcohol, ketone, aldehydes …) and with lifetimes ranging from minutes to months. They can be released directly into the atmosphere by anthropogenic (vehicular exhausts, evaporation of




gasoline, solvents use, natural gas emissions, industrial process) and natural sources (vegetation, ocean, etc.). Even though biogenic emissions of VOC are more important than anthropogenic emissions at a global scale (Finlayson-Pitts and Pitts 2000;

Goldstein and Galbally 2007; Müller 1992), the latter are the most dominant in urban areas.

Once released into the atmosphere primary VOC undergo chemical transformations (oxidations) due mainly to the presence of OH radical during the day. This yields to the formation of secondary Oxygenated VOC (Atkinson 2000; Goldstein and Galbally 2007), tropospheric ozone (Seinfeld and Pandis, 1998) and secondary organic aerosols (SOA) (Koppmann 2007; Hester and Harrison 1995; Fuzzi et al., 2006).

Some areas on the earth are more impacted by air pollution than others, which is the case in the Eastern Mediterranean Basin (EMB). This region is affected by both particulate and gaseous pollutants. The EMB undergoes environmental and anthropogenic pressures. Future decadal projections point to the EMB as a possible "hotspot" of poor air quality with a gradual and continual increase in temperature (Pozzer et al., 2012; Lelieveld et al., 2012). In this region, fast urbanization, high population density, industrial activities and on-road transport emissions, enhance the accumulation of anthropogenic

emissions. Natural emissions and climatic conditions (i.e. intense solar radiations, rare precipitations, and poor ventilation) also favor the photochemical processes. Therefore, the characterization and quantification of present and future emissions in the EMB are crucial for the understanding and management of atmospheric pollution and climate change at local and regional scales.

In this context, the project TRANSEMED (TRANSport, Emissions and Mitigation in the East Mediterranean,

(http://charmex.lsce.ipsl.fr/index.php/sister-projects/transemed.html) associated to the international project ChArMEx (Chemistry-Aerosol Mediterranean Experiment, http://charmex.lsce.ipsl.fr/) aims to assess the state of atmospheric pollution due to anthropogenic activities in the East Mediterranean basin urban areas. Up to now VOCs and their sources have been extensively characterized by implementing receptor oriented approaches in Beirut, Lebanon (Salameh et al., 2016) and Athens, Greece (Panopoulou et al., 2018). Field work and source-receptor analyses in Beirut (Salameh et al., 2014; Salameh et al.,

2016) have provided the first observational constraints to evaluate local and regional emission inventories in the EMB. A large underestimation up to a factor of 10 by the emission inventories was found suggesting that anthropogenic VOC emissions could be much higher than expected in the EMB.

The megacities of Istanbul (15 million inhabitants, this work) and Cairo are the next target urban areas. Istanbul has experienced rapid growth in urbanization and industrialization (Markakis et al., 2012). The region undergoes very dense

industrial activities with approximatively 37 % of industrial activities from the textile, 30 % from metal , 21 % from chemical industry ; 5 % from food and 7 % from the other industries (Markakis et al., 2012). Most of the experimental studies have focused on particulate matter (PM) and ozone in Istanbul in order to evaluate factors controlling their distribution. According to these studies, the major source of PM is of anthropogenic origin : refuse incineration, fossil fuel burning, traffic, mineral industries and marine salt (Koçak et al., 2011; Yatkin and Bayram, 2008). Markakis et al. (2012) determined that $PM_{10}$

originates mainly from industrial sources while fine particles ($PM_{2.5}$) are as much emitted by industry and transport. Beyond sources, the geographical location of Istanbul, (Black Sea in the North and the Marmara Sea in the South) produces surface



heating differences leading to different meteorological conditions that play an important role in the transport pollutants, especially of ozone (İm et al., 2008). Over the last two decades the only VOC measurements have been reported for other cities in Turkey by the use of off-line sampling and GC-FID or GC-MS (gas chromatography-mass spectrometry) analysis (Yurdakul

et al., 2013; 2018; Muezzinoglu et al., 2001; Elbir et al., 2007; Demir et al., 2011).

More recently VOCs source apportionment have been performed in Izmir, Bursa and Ankara (Elbir et al., 2007; Yurdakul et al., 2013; 2018) on a VOC dataset usually including alkanes, alkenes and aromatic compounds. Except for Izmir, traffic was not the dominant source for urban VOC in any of these source apportionment studies and industrial emissions drive the VOC distribution. This is in contrast with what is usually found in mid latitudes cities as well as in Beirut where traffic exhaust and

gasoline evaporation seem to dominate (Salameh et al., 2016).

The objective of this work is to analyze the VOC concentration levels, their variability and to apportion their emission sources in the megacity of Istanbul in order to evaluate emission inventories downscaled to the megacity. According to the local emission inventory by Markakis et al. (2012), the main emission sources of VOC in Istanbul would come from traffic (45%), solvent use (30%), waste treatment (20%) while less than 1% originates from industrial processes. Our methodology is based

on the Positive Matrix Factorization model (PMF) which can be applied without prior knowledge of the source compositions. Moreover the model is constrained to non-negative species concentrations and source contribution. VOC source apportionment by PMF have been already successfully conducted in many urban areas: Los Angeles (Brown et al., 2007), Paris (Gaimoz et al., 2011; Baudic et al. 2016), Beirut in Lebanon (Salameh et al., 2016), Zurich, Switzerland (Lanz et al., 2008). A large set of speciated VOC were continuously collected during a 2-week intensive field campaign in September 2014 at an urban site in

Istanbul, along with ambient measurements at various locations in the megacity.

## 2 Methods

### 2.1 Domain and measuring sites

The megacity of Istanbul has a unique geographical location spanning on two continents, Europe and Asia (Anatolia)

(Figure 1). Due to its location in a transitional zone, the city experiences Mediterranean, humid subtropical and oceanic climates with warm/dry summers and cold/wet winters. The Black Sea in the North and the Marmara Sea (Figure 1) in the South produce a heat gradient at the surface leading to meteorological conditions likely to play a major role in atmospheric pollution (İm et al., 2008). The wind roses observed during the campaign period are reported in Figure 1 and are typical of summertime conditions. They show that the wind direction is mainly North-East (NE).

The urban site (labelled as supersite in Figure 1) is located along the Barbaros Boulevard (Blvd) in the district of Besiktas in Istanbul, Turkey (41°02′33″N, 29°00′26″E) on the European shore of the Bosphorus strait. Barbaros Blvd is a high traffic density street, which is a major route between the city center (a.k.a. "Historical Peninsula") and the Bosphorus Bridge that connects Asia and Europe. The Besiktas district is also expected to be impacted by the mixture of major anthropogenic



emissions (Markakis et al., 2012). Note that the sampling site was 500 m away from the Besiktas Pier (Bsks-shore on Figure
1) and the Bosphorus Strait 4-km away from the Haydarpasa Port. Dense industrial areas like Ikitelli area are located in the
southwestern part of the city 20-km away from Besiktas. Trace gases measurements including VOC were conducted at the
supersite from September 14$^{th}$ to September 30$^{th}$ of 2014 (during the non-heating season). In parallel, punctual VOC sampling
at four other locations have been performed to assess the spatial variability of VOC composition and to support the
interpretation of PMF factors. Punctual sampling included one residential area in the district of Kagithane (Kag –09/29/2014
and 09/30/2014), one roadside site on the Barbaros Blvd (Bskts – 09/24/2014), two sea-shore sites at the Besiktas Pier (Bskts-
shore 09/26/2014), and on the Galata Bridge (Gal-shore 09/29/2014). Local ferries that connect the European and Asian sides
continuously travel to the Besiktas pier with few minutes parking time at berth. The Galata Bridge spans the Golden Horn in
Istanbul with a roadway on its upper part. For data analysis the Greater Municipality of Istanbul provided air quality data (CO,
$NO_x$ and $SO_2$) from its operated Bskts site (Figure 1), and meteorological data were obtained from Turkish State Meteorological
Office (wind speed, wind direction, temperature, relative humidity and ambient pressure) from its network (Sariyer and Florya
stations on Figure 1).

## 2.2 VOC instrumentation

The on-line instrumentation for VOC at the supersite included an AIRMOVOC GC-FID (Gas Chromatograph Flame Ionisation
Detector, Chromatotec®) and a high sensitivity PTR-MS (Proton Transfer Reaction Mass Spectrometer, Ionicon®) at 30-min
and 5-min time resolution, respectively. Principle, performances and operation conditions of both instruments have been
described elsewhere (Gaimoz et al., 2011; Borbon et al., 2013 ; Ait-Helal et al., 2014).

Ambient air sampling was performed at the height of 2 m a.g.l. Air was pulled through two independent 3-m Teflon lines
(PFA, ¼" outside diameter) towards the instruments. During the TRANSEMED campaign, the GC-FID measured 15 VOC
from C4 to C8 (alkanes, alkenes, aromatics). VOC-free zero air and a certified ppb level gaseous standard from NPL (National
Physical Laboratory, UK) at 4 ppb ± 0.8 ppb were injected every three-days. Eleven protonated masses were monitored with
the PTR-MS. The background signal was determined every 2 days for 30 min by passing air through a catalytic converter
containing platinum-coated pellets heated to 320°C. The drift pressure was maintained at 2.20 mbars and the drift voltage at
600 V. The primary $H_3O^+$ ion counts at m/z 21 ranged between $0.9 \times 10^7$–$1.4 \times 10^7$ cps with a contribution from the monitored
first water cluster at m/z 37 < 5 %. Two multi-point calibrations were performed using a Gas Calibration Unit (GCU), and a
standard mixture of 17 species (Ionimed Analytik GmbH, Innsbruck, Austria), (Singer et al., 2007) before and after the
campaign over a 0.1-20 ppb range ; the linearity was higher than 0.99 ($r^2$). The species used for the calibration were methanol
(contributing to m/z 33), acetaldehyde (m/z 45), acetone (m/z 59), isoprene (m/z 69), crotonaldehyde (m/z 71), 2-butanone
(m/z 73), benzene (m/z 79), toluene (m/z 93) and α-pinene (m/z 137). In addition, five regular 5-ppb control points were
carried out during the campaign. Except m/z 33 (17 %), the standard deviation of the calibration coefficients lay between 3 %
(m/z 93) and 9 % (m/z 42). The NPL standard was used to cross-check the quality of the calibration and to perform regular
one-point calibration control for isoprene and C6-C9 aromatics (4.0 ± 0.8 ppb). A relative difference of less than 10 % was





found between both standards. The calibration factor for all major VOC (the slope of the mixing ratio with respect to product ion signal normalized to $H_3O+$ and $H_3O+H_2O$) ranged from 2.54 (m/z 137) to 19.0 (m/z 59) counts per seconds. Detection limits were taken as $2\sigma$ of the background signal divided by the normalized calibration factor. Detection limits lay between

50 ppt (e.g. monoterpenes) and 790 ppt (methanol). Finally eleven protonated target masses have been monitored here: methanol (m/z 33.0), acetonitrile (m/z 42.0), acetaldehyde (m/z 45.0), acetone (m/z 59.0), methyl vinyl ketone (MVK) and methacrolein (MACR) (m/z 71.0), benzene (m/z 79.0), toluene (m/z 93.0) and C8-aromatics (m/z 107.0), C9-aromatics (m/z 121.0) and terpenes (m/z 137.0). While Yuan et al. (2017) reports interferences in its review paper (Yuan et al., 2017) for some of these protonated masses they can be excluded in high-NOx environment.

Off line instrumentation, which included sorbent tubes and canisters, provides a lighter set-up to describe emission source composition and the spatial variability of VOC composition to support the PMF analysis. The instrumentation was deployed at the supersite and at the four locations reported on Figure 1 (VOC label). Off-line measurements of C5 to C16 NMHCs (alkanes, aromatics, alkenes, aldehydes) were performed onto multibed sorbent tubes of Carbopack B & C (Sigma–Aldrich Chimie S.a.r.l., St Quentin Fallavier, France), at a 200 mL min$^{-1}$ flow rate for 2 h using a flow-controlled pump from GilAir.

Samples were first thermodesorbed and then analysed by TD-GC-FID/MS within one month after the campaign. The sampling and analysis method are detailed elsewhere (Detournay et al., 2011). Air samples were also collected by withdrawing air, for 2–3 min, into preevacuated 6-L stainless steel canisters (14 samples) through a stainless steel line equipped with a filter (pore diameter = 2 μm) installed at the head of the inlet. Prior to sampling, all canisters were cleaned at least five times by repeatedly filling and evacuating zero air. Tubes and canisters were sent back to the laboratory (Salameh et al., 2014). The frequency of

off-line samplings is reported on Figure S1 at the various site reported in Figure 1.

The compounds commonly measured by on-line and off-line techniques, namely aromatics, isoprene, pentanes and terpenes were used to cross-check the quality of the results at the supersite on September 26$^{th}$ and 29$^{th}$. The comparison is provided in Figure S2 in the Supplement Material. The variability between PTRMS and AIRMOVOC is highly consistent ($r^2 > 0.85$) and the differences in concentrations do not exceed ±20 %. While the number of off-line samples is limited (5 samples), the values

are consistent. The pentanes, terpenes and C9-aromatics collected by tubes or canisters are also compared to AIRMOVOC (Figure S3). Most of the data compare well at ±20 % with few exceptions which do not exceed ±50 %. Finally, the correlation between PTRMS and AIRMOVOC is surprisingly weak for isoprene. Despite some interferences like furans cannot be excluded for m/z 69 with PTR-MS (Yuan et al., 2017), the good correlation between m/z 69 from PTRMS with ambient temperature led to use the PTRMS data for isoprene.

**2.3 Other instrumentation**

At the supersite, other trace gases like carbon monoxide (CO) and $NO_x$ ($NO+NO_2$) have been performed on a 1-minute basis. $NO_x$ was monitored by the Thermo Scientific model TEI 42I instrument model, which is based on chemiluminescence and $NO_2$-to-NO conversion by a heated molybdenum converter. The $NO_x$ analyzer was installed after September 25$^{th}$ at the supersite. CO was monitored by the Horiba APMA-370 instrument model which is based on Non-Dispersive Infra-Red (NDIR)



technique. Basic meteorological parameters (wind speed and direction, temperature, relative humidity, and atmospheric pressure) were measured on a 1 min basis. Air quality data with additional sulfur dioxide ($SO_2$) and meteorological data from other stations operated by the Greater Istanbul Municipality have been also used to test the consistency of our data and to support PMF interpretation (see Figure 1and section 1.1).

In addition to gaseous and meteorological parameters, particulate matter samples were also collected at the site. Partisol $PM_{2.5}$

sequential sampler (Thermo Scientific, USA) was deployed (24 hours between 28th of August and 13th of November 2014 and every 6 hours until 28th of January 2015) and collected samples were analyzed in terms of metals by ICPMS. Moreover, Tecora $PM_{10}$ sampler (Italy) was used to collect daily $PM_{10}$ samples (30th of August 2014-3rd of February 2015) and collected samples were analyzed in terms of EC and OC by means of Sunset Lab (Oregon, USA) thermal optical analyzer, molecular organic markers by using GCMS (Varian CP 3800 GC equipped with a TR-5MS fused silica capillary column) and WSOC by

means of an organic carbon analyzer (TOC-VCSH, Shimadzu). The sampling time and availability of the data are reported in Figure S1 of the Supplement Material.

### 2.4 Positive Matrix Factorization

### 2.4.1 PMF description

The US EPA PMF 5.0 was used for VOC source apportionment. The method is described in detail in Paatero and Tapper,

1994; Paatero, 1997 and Paatero and Hopke, 2003.

The general principle of the model is as follow: any matrix X (input chemical data set matrix), can be decomposed in a factorial product of two matrixes G (source contribution) and F (source profile), and a residual part not explained by the model E. Equation 1 summarizes this principle in its matrix form:

$$x_{ij} = \sum_{k=1}^{p} g_{ik} f_{kj} + e_{ij} \qquad (1)$$

Where i is the number of observations, j the amount of the measured VOC species and k the number of factors.

The goal of the PMF is to find the corresponding non-negative matrixes that lead to the minimum value of Q.

$$Q = \sum_{i=1}^{n} \sum_{j=1}^{m} \frac{e_{ij}^2}{s_{ij}^2} = \sum_{i=1}^{n} \sum_{j=1}^{m} \left( \frac{x_{ij} - \sum_{k=1}^{p} g_{ik} f_{kj}}{s_{ij}} \right)^2 \qquad (2)$$

Where $f_{kj} \geq 0$ and $g_{kj} \geq 0$ and where n is the number of samples, m the number of considered species, and $s_{ij}$ an uncertainty estimates for the $j^{th}$ species measured in the $i^{th}$ sample.

### 2.4.2 Preparation of input data

Two input dataset (or matrixes) are required by the PMF: the first one contains the concentrations of the individual VOC and the second one contains the uncertainty associated with each concentration.





The VOC dataset combines data from both the PTRMS and the GC-FID. Data have been synchronized on the lower time resolution basis (30 min). The final chemical database used for this study comprises a selection of 23 hydrocarbons and masses

divided into 7 compound families: alkanes (isobutane, n-butane, n-hexane, n-heptane, isopentane, n-pentane and 2-methyl-pentane), alkenes (1-pentene, isoprene, and 1,3-butadiene), aromatics (ethylbenzene, benzene, toluene, (m+p)-xylenes, o-xylene, and C9-aromatics), carbonyls (methylethylketone (MEK), methacroleine+methylvinylketone (MACR+MVK), acetaldehyde, and acetone), alcohol (methanol), nitrile (acetonitrile), and terpenes. Alkanes and alkenes were measured by the GCFID while C9 aromatic, carbonyls, alcohol, nitrile, and terpenes were measured by the PTRMS. Isoprene, benzene, toluene,

and C8 aromatics were measured by both instruments.

Since the PMF does not accept missing values, missing data must be replaced. The percentage of missing values ranges from 8 to 79 % for species measured by the GC-FID. Butanes (iso/n), pentanes (iso/n), and (m+p)-xylenes have the lowest missing values percentage (ranging from 8 to 12 %) while o-xylene (30 %), 2-methyl-pentane (32 %), %), ethylbenzene (42 %), 1-pentene (62 %), n-hexane (66 %), n-heptane (72 %), and 1,3-butadiene (79 %) have the highest percentage of missing values.

There were only 2 missing values (0.33 %) for species measured by the PTRMS.

For those species with a proportion of missing values below 40 %, missing data were replaced by a linear interpolation. For species with a proportion of missing data exceeding 40 %, each missing data point was substituted with the median concentration over all the measurement period. All the concentrations were above the detection limit.

The uncertainty $\sigma_{ij}$ associated to each concentration (Equation 3) is determined using the method developed within the

ACTRIS (Aerosol, Cloud and Trace Gases Research Infrastructure) network (Hoerger et al., 2015) and used in (Salameh et al., 2016).

$$\sigma_{ij} = \sqrt{precision^2 + accuracy^2} \qquad (3)$$

This uncertainty considers the different sources of uncertainty affecting the precision and the accuracy terms. The precision is associated with the detection limit of the instrument, the repeatability of the measurement, while the accuracy includes the

uncertainty of the calibration standards and the dilution when needed. Further information about the uncertainties calculation are found in the section 1 of the supplement material.

**2.4.3 Determination of the optimal solution**

The objective of the PMF is to determine the optimal number of factors (p) based on several statistical criteria. Several base runs were performed with a different number of factors from 2 to 12. Specific parameters were then used to determine the

appropriate p value such as Q (residual sum of squares), IM (maximum individual Column mean), IS (maximum individual column standard deviation) defined by Lee and al. (1999) and $R^2$ (indicator of the degree of correlation between predicted and observed concentrations). Q, IM , IS and $R^2$ are then plotted against the number of factor (from 2 to 12) (Salameh et al., 2016). The number of chosen factors corresponds to a significant decrease of Q, IM, and IS. In our study, an optimal solution of 5 factors was retained.





In order to ensure the robustness of the solution, a Fpeak value of -1 was set by considering the highest mean ratio of the total contribution vs the model as well as the numbers of independent factors ( Salameh et al., 2016).

## 3 Results and discussion

### 3.1 Meteorological conditions and air mass origin

Meteorological parameters are reported in Figure 2 and air mass origin from the Lagrangian particle dispersion FLEXPART

model are reported in Figure S4 of the supplement material. FLEXPART is widely used to stimulate atmospheric transport. It gives information about long range and mesoscale dispersion of air pollutants such as air mass origin (back trajectories) (Brioude et al., 2013). FLEXPART was driven by ECMWF analysis (at 00, 12 UTC) and their 3 hourly forecast fields from the operational European Centre for Medium Range Weather Forecasts - Integrated Forecast System (ECMWF-IFS). To compute the FLEXPART trajectories, the ECMWF meteorological fields were retrieved at 0.25° resolution and 91 vertical

levels. Three mains periods were observed separated by short transition periods (figure S4):

Based on in situ parameters and FLEXPART simulations, the meteorological situations can be divided into four types of periods:

-  Periods 1 and 3 from 09/15 to 09/21 and from 09/28 to 09/30 respectively with air masses coming from Black Sea and Russia and/or Ukraine and Northern Europe (marine influence). During period 1, temperature and relative

humidity were characterized by a clear and opposite diurnal cycle while no cycle is depicted during period 3. During both periods relative humidiy is high with two rain events (09/18 and 09/28). Relative humidity is high during the night (80 %) while temperature is high during the day (from 14.7 to 27.6°C with an average of 21.2 °C. Wind speed are the highest high (>3 m.s$^{-1}$ in Besiktas and up to 12 m.s$^{-1}$ at Sariyer) with a northern win direction.

-  Period 2 from 09/23 to 09/26 with air masses coming from Eastern and Northern Europe (continental influence).

Locally wind direction is variable. Temperature and relative humidity were characterized by the same diurnal cycles as in Period 1. Temperature decreased, ranging between 11.8 and 25.4 °C with an average of 17.9 °C. Lower wind speeds were recorded (3.1 m.s$^{-1}$ in Besiktas and up to 6 m.s$^{-1}$ in Sariyer). Wind direction varied between WSW (West-South-West) and SSE (South-South-East), and between North and ENE (East-North-East).

-  A first transition period between period 1 and 2 (09/21 to 09/23) showed a wind direction shift towards the southern

sector (Marmara Sea). Wind speed recorded the lowest values (<1.3 m.s$^{-1}$ in Besiktas and <5 m.s$^{-1}$ in Sariyer). It rained on 09/23. A second transition period between period 2 and 3 (from 09/26 to 09/28) showed a wind direction shift toward the NNE (North-North-East) (supplement material Figure S4). On 09/06, the temperature went up to 26 °C. It rained from the night of 09/26 until the end of the period during which temperature decreased down to 14.7 °C. This period was also characterized by higher wind speed (up to 9 m.s$^{-1}$ in Sariyer and 2.7 m.s$^{-1}$ in Besiktas) coming

from ESE (East-South-East) and North.



### 3.2 VOC distribution

#### 3.2.1 Ambient concentration levels and composition

Observed VOC at the supersite of Besiktas are representative of the ones usually encountered at urban background sites. Biogenic compounds are isoprene and terpenes while anthropogenic compounds include both primary (alkanes, alkenes,

aromatic compounds) and secondary compounds (oxygenated VOC).

Statistics on the VOC concentrations measured on-line by GC-FID and PTR-MS at the supersite of Besiktas are reported in Table 1. VOC show a high temporal variability with maxima reaching several tens of ppb (isopentane, ethylbenzene, methanol, and acetone). Most of VOC median concentrations are below 1 ppb except for n-butane (1.48 ppb), toluene (1.25 ppb) and some oxygenated like acetaldehyde (1.39 ppb) and acetone (2.33 ppb). The average composition of VOC is mainly composed

of OVOC (0.12-4.92 ppb) which represent 43.9 % of the total VOC (TVOC) observed mixing ratio, followed by alkanes (0.21-2.00 ppb; 26.33 %), aromatic compounds (0.26-2.27 ppb; 20.66 %), alkenes (0.19-0.68 ppb; 4.81 %), terpenes (3.44 %) and acetonitrile (0.84 %). Both OVOC and alkanes contribute up to 70.2 % of the TVOC concentrations. Methanol (22.40 %, 4.92 ppb on average) is the main oxygenated compound measured in this study, followed by acetone (11 %, 2.63 ppb). N-butane (9.11 %, 2.00 ppb) and isopentane (6.43 %, 1.41 ppb) are the major alkanes. Toluene is the most abundant aromatic compound

(50 %, 2.27 ppb). This is the case in most of the VOC studies made in Istanbul (Demir et al., 2011; Kuntasal et al., 2013; Muezzinoglu et al., 2001; Bozkurt et al., 2018; Elbir et al., 2007).

Levels of alkanes, some alkenes and aromatics are compared to other European megacities (Paris and London) at both urban and traffic site as well as at a suburban site in Beirut during summer (Figure S5 in the supplement material). Consistency in urban hydrocarbon composition worldwide has been already observed (Borbon et al., 2002; von Schneidemesser et al., 2010;

Dominutti et al., 2016). Once again, the composition in hydrocarbons in Istanbul is consistent with the ones between all the cities. Beirut has the highest concentrations of n-butane, isopentane and 2-methyl-pentane. Higher concentrations in toluene, (m+p)-xylenes were in Paris, Beirut and Istanbul. This would suggest the traffic influence on the composition in hydrocarbons in Istanbul (see also section 3.3 on PMF).

Off-line VOC concentrations collected with tubes are reported in Table S6 in the Supplement material at various locations

including the Besiktas supersite. While the number of samples is limited in time, off-line measurements provide a general picture of a wider spectrum of VOC that are not measured at the supersite like lighter hydrocarbons (C2-C3 alkanes and alkenes), C6-C11 carbonyls, speciated terpenes (camphene, limonene, β-pinene) and C12-C16 IVOC. Therefore, these dataset allow to address the spatial variability of VOC concentrations and composition within the megacity. The comparison of concentrations between different areas is limited because of the different sampling times. Therefore, the analysis only focus

on VOC relative composition (Figure 3). The composition is variable across the megacity for the aliphatic fraction of high and intermediate volatility hydrocarbons (C2-C16). As expected, one VOC fingerprint by canister at the supersite (29/09 12:05 sample) is like the ones derived from the roadway side measurements highlighting the influence of road transport emissions. Interestingly, the other VOC fingerprints from sorbent tubes at the supersite are different from the ones at the roadway side



with a higher proportion of IVOC. The other VOC fingerprint by canister at the supersite (26/09 10:31 sample) is rather similar
to the one from the seashore sample in Galata. In the same way the VOC fingerprints at the supersite derived from tube samples
are rather like the seashore ones (Besiktas). For both of them, the proportion of IVOC is significant (from 15 % to 40 % in
weight). While light VOC are expected to be of minor importance when considering ship emissions, the higher presence of
heavier organics is however expected as observed for alkanes by Xiao et al. (2018) in ship exhaust at berth. The VOC
fingerprint comparison would thus suggest not only the impact of road traffic emissions on their composition but also the
potential impact of local ship traffic emissions.

### 3.2.2 Time series and diurnal variations

The variability of VOC concentrations is driven by several factors: emissions (anthropogenic or biogenic), photochemical
reactions (especially with the OH radical during the day and ozone and nitrates at night for alkenes) and the dynamic of the
atmosphere (incl. dilution due to the height of the boundary layer) (Filella and Peñuelas, 2006). The time series of air quality
trace gases ($NO_x$ and CO) and representative VOC are reported in Figure 4. The meteorological periods 1, 2 and 3 are also
indicated. Because $NO_x$ at the super site were only measured from 09/25 to 09/30, data from the air quality station in Besiktas
were used (see Figure 1).

$NO_x$ and CO exhibit different patterns. While $NO_x$ show a clear diurnal cycle with maximum concentrations at midday as
expected from traffic-related compounds, CO does not show the same cycle. Conversely, VOC time series exhibit different
patterns from those of $NO_x$ and CO. Anthropogenic VOC do not show any clear diurnal cycle. On the contrary, isoprene and
its oxidation products (MACR+MVK) covariate most of the time. They usually show their typical diurnal profiles with higher
concentrations during the warmest days and at midday due to biogenic emission processes. Their significant correlation with
temperature (r = 0.7) implies the emission from biogenic sources. Around the Besiktas site, 49.5 % of the vegetation is occupied
by hardwood and hardwood mix trees while only 6 % is occupied by softwood and hardwood mix trees. Pinus, which are
terpene emitter, represent up to 33.2 % of the overall vegetation in Istanbul. Quercus (ioprene emitter) only occupy 7.7 % of
the total vegetation coverage (personal communication from Ministry of Forestry). While not systematic, highest
concentrations are observed during the night for anthropogenic VOC especially during period 2. One cause are the very low
wind speeds at night especially during period 2 (Figure 2), which would reinforce the accumulation of pollutants. Under marine
influence (periods 1 and 3), VOC concentrations are the lowest, especially during period 3, which is characterized by rainy
days (September 27th and 28th), high wind speed and colder temperatures (Figure 4). These conditions favor atmospheric
dispersion. During transition periods and under continental influence (period 2), VOC concentrations exhibit a strong day-by-
day variability with episodic nocturnal peaks especially on September 25th and 26th. While concentration peaks are not always
concomitant between VOC, they occur under south and southwestern wind regimes which are unusual according to Figure 1.
This points out the potential influence of industrial and port activities. For instance, maximum concentrations of butanes
occurred during the period of the marine-continental regime shift with well-established southwestern wind regime on 09/22,
09/23 and on 09/26 at the end of the day. Maximum concentrations of pentanes occurred during the night of 09/26 to 09/27



like for aromatics (e.g. benzene) (Figure 4). Some VOC show greater background concentration levels during this period like acetone and isobutane. Finally, time series would suggest the influence of multiple sources other than traffic on VOC concentrations, likely industrial and/or port activities, at the supersite. After road transport, cargo shipping is a second highest

contributor to $NO_x$ levels according to the local/regional inventory (Markakis et al., 2012). Contrary to VOC, $NO_x$ does not show any concentration peak despite its high-frequency variability. Such variability of these time series raises the question of the role of traffic on VOC distribution. The nature and importance of VOC sources will be further investigated in the PMF section (section 3.3).

Taking into consideration time series variability, diurnal variations have been splitted into periods 1 and 3 and period 2 for

selected VOC as well as two combustion derived trace gases ($NO_x$ and CO). Diurnal profiles of atmospheric concentrations are reported in Figure 5. Local traffic counts for road transport (personal communication from Istanbul Municipality for fall 2014) and ship (https://www.marinetraffic.com/en/ais/details/ports/724/Turkey_port:ISTANBUL) are also reported in Figure S7 in the supplement material. Maritime traffic is mostly for passenger shipping (58.02%) against 16% for cargo shipping 16%. The diurnal profiles of ship and road traffic counts are similar.

Generally, concentrations during period 2 are higher than the ones during periods 1 and 3 and show different diurnal patterns.

The profile of $NO_x$ is consistent with the one of traffic counts (Figure S7 of the supplement material). $NO_x$ exhibits higher concentrations during the day and lower concentrations at night for both period with a morning peak (7:30-8:30) and one early evening peak from 17:30 (Figure 5.a). This is typical of one of traffic emitted compounds with morning and evening rush-hour peaks as observed in many other urban areas like Paris (Baudic et al., 2016) or Beirut in Eastern Mediterranean (Salameh et

al., 2016).

$NO_2/NO_x$ ratio in this study fluctuate between 0.34 to 0.93 with an average and median value of 0.53 and 0.55 respectively. These values are very high compared to what is usually found in the literature (Grice et al., 2009; Kousoulidou et al., 2008; Keuken et al., 2012) which is mosly low and below (50 %). However higher values of $NO_2/NOx$ ratio can be found in diesel passenger cars (Grice et al., 2009. Vestreng et al., 2009) and vans (Kousoulidou et al., 2008).

Surprisingly CO diurnal profile is different from the one of $NO_x$. CO is characterized by a double peak: one in the morning (8:30) and the other in the middle of the day followed by a constant concentration during the rest of the day (Figure 5.b) and higher concentrations at night. Similarly to CO, VOCs show different profiles from the one of NOx. Under marine influence (periods 1 and 3), primary anthropogenic VOC (ie. benzene, alkanes and other aromatics) almost exhibit a constant profile. Under continental influence (period 2), they show higher concentration at night. For instance, benzene (Figure 5.d) and

isopentane (Figure 5.f) concentrations increase by four-fold compared to the levels under marine influence. During the day the concentration levels are the same as during periods 1 and 3. The profiles of anthropogenic compounds point out the complex interaction between local and regional emissions and dynamics with high wind speed during the day. Period 2 points out the





influence of VOC emissions other than traffic (no effect on $NO_x$) at night. While the influence of traffic emissions on CO and VOC cannot be excluded; it seems that their emission level is not high enough to counteract the dispersion effect during the day unlike $NO_x$. This will be further investigated in the PMF analysis.

Isoprene concentrations increase immediately at sunrise and decrease at sunset which indicates its well-known biogenic origin (Figure 5.g) which is light and temperature dependent. Indeed, the site is surrounded by harwood and harwood mix trees composed of pinus (terpenes emitters) and quercus (isoprene's emitters) the most abundant ones. Like other anthropogenic VOCs, isoprene concentrations increase at night during period 2. Provided some interferences like furans could contribute to isoprene signal by PTRMS measurements (Yuan et al., 2017), this would suggest an anthropogenic origin for isoprene. During period 1 and 3 to their major primary precursor (isoprene), MACR+MVK also show high concentrations at night during period 2 like other alkanes and aromatics, suggesting their potential anthropogenic influence. During periods 1 and 3 (marine influence), MACR+MVK concentrations follow the same general pattern as of isoprene's.

With relatively long atmospheric lifetime, ($\approx$ 68 days), acetone's concentration is quite constant throughout the day within period 1, a peak in the middle of the day and lower concentrations during the night for period 2 (Figure 5.c). The peak in the middle suggests the presence of a secondary source. Acetone can have both primary and secondary source (Goldstein and Schade, 2000; Macdonald and Fall, 1993). Methanol and MEK have the same general pattern as for acetone during both periods without the peak in the middle of the day suggesting that they might have the same emission source.

## 3.3 PMF results

### 3.3.1 Factor identification

PMF reference run has been performed by removing the period during which there were no GC-FID data (night from 09/24 to 09/25). In addition, these data set have been chosen as PMF reference run because of the higher correlation between observed and reconstructed data by the PMF model (see also section 3.2). A good correlation ($r^2 = 0.97$) between total VOC reconstructed and measured concentrations was obtained. For most compounds the variability is well reconstructed with $r^2$ usually higher than 0.70. Poorer correlation was found for alkenes (1-pentene ($r^2 = 0.55$), 1,3-butadiene ($r^2 = 0.22$) and isoprene ($r^2 = 0.57$)) as well as for n-hexane ($r^2 = 0.09$), MEK ($r^2 = 0.41$) and acetaldehyde ($r^2 = 0.32$).

The PMF output uncertainties were estimated by three models: the DISP model (base model displacement error estimation), the BS model (base model bootstrap error estimation) and the DISP+ BS model. Further information for the estimation of model prediction uncertainties can be found in Norris et al. (2014) and Paatero et al. (2014). The DISP results of the PMF run show that the 5-factor solution is stable and sufficiently robust to be used because no swaps occurred. All the factors were well reproduced through the BS technique at 100 % for factor 1, 96 % for factor 2, 100 % for factor 3, 99 % for the factor 4 and 100 % for factor 5; there were not any unmapped run. The DISP+BS model shows that the solution is well constrained and stable.



The factor profiles have been analyzed using the variability of their contribution together with several external variables (NO$_x$,
SO$_2$, CO, meteorological parameters, emission profiles). PMF factors are displayed Figure 6. The time series and diurnal
profiles of their contributions are displayed in Figures 7 and 8, respectively.

**Factor 1: Toluene from solvent use**

The speciation profile of factor 1 exhibits high concentrations of toluene with 57 % of its variability explained by this factor.
There is also a small contribution of acetone (18 %), MEK (18 %) and xylenes (17 %). While toluene and xylenes are related
to traffic emissions, this factor does not correlate well with any traffic tracer gases (r = 0.33 for NO$_x$,) and combustion trace
gases in general (r=-0.03 for CO and 0.13 for SO$_2$). Moreover toluene also contributes to the Road transport factor (factor 5)
but to a lesser extent. The time series are highly variable with erratic peaks regardless of the time of the day especially during
period 2 and to a lesser extent period 1 (Figure 7). The average diurnal profile looks like constant but the relative standard
deviation is ±100%. Sources related to solvent use are among the expected non-combustive sources for toluene (Baudic et al.,
2016; Gaimoz et al., 2011;  (Brocco et al., 1997; Na and Kim, 2001). In Turkey, toluene was already found in gasoline vehicle,
solvent and industrial emissions (Bozkurt et al., 2018; Yurdakul et al.,2013; Demir et al., 2011).

Toluene/benzene ratio (T/B) is used as an indicator of non-traffic source influence (Elbir et al., 2007; Lee et al., 2002; Yurdakul
et al.,2013). Low T/B ratio (≤ 2-3) indicates the influence of traffic emissions on measured VOC concentrations (Gelencsér et
al.,1997; Heeb et al., 2000; Muezzinoglu et al., 2001; Brocco et al., 1997) whereas T/B ratios ≥ 2-3 suggests the influence of
other sources than traffic (such as solvent evaporation or industrial sources).  The T/B ratio for this study is between 0.4 (with
only 4 points below 2) and 48.6 (Only 1 point above 29). Only 5.8 % of the ratios were between 2 and 3, 48 % were between
3 and 6 while 45 % were above 6 with 34 % between 6 and 10. This strongly suggests the influence of sources of toluene other
than traffic. High value of T/B ratio is mostly found at industrial sites (Pekey and Hande, 2011). The median and mean value
of T/B in this experiment are respectively 5.6 and 6.7 which can also indicate gasoline related emissions (Batterman et al.,
2006). However the absence of other unburned fuel compounds like pentanes excludes this source. This factor represents
14.2 % of the total contribution.

**Factor 2: Biogenic terpenes**

Factor 2 exhibits high contribution of terpenes with more than 73 % of their variability explained by this factor (Figure 6).
Terpenes are known as tracers of biogenic emissions (Kesselmeier and Staudt, 1999). Isoprene which is also a biogenic tracer
has only 5 % of its variability explained by this factor. Moreover, the diurnal profile of these two compounds show opposite
patterns  as it can be seen in Figures 5 and 8 which indicates that their biogenic emissions are controlled by different
environmental parameters: temperature for terpenes, light and temperature for isoprene (Fuentes et al., 2000). This factor
shows high concentrations and contribution during period 1 and the transition periods (Figures 7 and 2) probably due to higher
temperature while they are almost not significant during period 3. As expected, terpenes do not correlate with any combustion
related gases (r <0.14 for NO$_x$, CO and SO$_2$).

The diurnal profile of this factor is characterized by high concentrations at night and early morning (until 8:30) and low
concentrations during daytime (Figure 8). This type of profile has already been observed at a background site in Cyprus



(Debevec et al., 2017); in a forest of Abies Boriqii-regis in the Agrafa Mountains of north western Greece (Harrison et al., 2001) and at Castel Porziano near Rome, Italy (Kalabokas et al., 1997). In a shallower nocturnal boundary layer, low chemical
reactions together with persistent emissions lead to the enhancement of their nocturnal mixing ratios. Terpene's lifetime toward OH and ozone are 1.2 to 2.6 h and 5 min (for α-pinene) to 50 min (for camphene) respectively (Fuentes et al., 2000). Dilution processes of the boundary layer in addition to the higher reactivity of terpenes towards the OH radical and ozone could explain the decrease of their concentrations during the day.

This factor is called "biogenic terpenes" and represents 7.8 % of the total contribution.

**Factor 3: Natural gas evaporation**

Factor 3 is essentially composed of butanes (iso/n) with more than 97 % of their variability explained by this factor (Figure 6). Isobutane is a typical marker of fossil fuel evaporative source (Debevec et al., 2017; Na et al., 1998). This factor significantly explains the contributions of 1,3-butadiene (32 %), acetonitrile (23 %), acetaldehyde (24 %), MACR + MVK (21 %), C9-aromatics (22 %) and benzene (20 %). Iso/n butanes correlate poorly with CO (r = 0.09) and NO (r = 0.33). The alkanes/alkenes
ratio of this factor is high (55), which points out the evaporative source of this factor (Salameh et al., 2014). At the same time, the pentanes (n/iso) and the other aromatic compounds are not well represented in factor 3. This suggests that butanes evaporation is not related to evaporation from storage, extraction and distribution of gasoline but rather to natural gas evaporation.

The diurnal profile of factor 3 is characterized by an increase in concentration at night and constant concentration from 10:00
to 18:00 with a slight peak in the middle of the day and in the evening (Figure 8). This points out a source linked to the use of natural gas as energy source, especially for cooking at lunch time with the proximity of many restaurants near the measurement site. This type of profile has already been observed in Paris (Baudic et al., 2016). The time series are characterized by several peaks during period 2, which corresponds to the marine transition by the south-southwest of Istanbul and the Marmara Sea (Figure 7). It is noteworthy the presence of a power plant with a capacity of 1350 MW located in the southwest of Ambarli
that uses natural gas as fuel and which can also contributes to the butane loads . This factor does not depend on temperature or on other trace gas (NO$_x$, CO and SO$_2$). The average relative contribution of natural gas factor is 25.9 %.

**Factor 4: Mixed diurnal regional emissions**

Factor 4 has a significant contribution of several primary and secondary as well as biogenic and anthropogenic species (Figure 7). N-hexane is the most dominant species (81.6 % of its variability is explained by this factor). However, knowing that this
compound has more than 70 % of missing values, the analysis related to this compound should be done carefully. The great contribution of isoprene (66 %) in factor 4 points out its biogenic emissions. Biogenic emissions of isoprene are directly related to temperature as well as solar radiation (Steiner and Goldstein, 2007; Owen et al., 1997; Geron et al., 2000). Note that this factor correlates well with the ambient temperature (r = 0.70). Despite the percentage of missing values, the biogenic contribution on this factor can also be explained by the presence of 1,3-butadiene and 1-pentene likewise emitted by plants
(Goldstein et al., 1996). The factor 4 is also characterized by the presence of oxygenated compounds such as isoprene oxidation products like MACR+MVK (54 %), acetaldehyde (66 %), acetone (57 %), methanol (59 %) and MEK (59 %,). These species





are formed by the oxidation of primary biogenic hydrocarbons. However, these oxygenated species can also have primary both anthropogenic and biogenic sources ( Yáñez-Serrano et al., 2016; Millet et al., 2010; Goldstein and Schade, 2000; Singh, 2004; Schade et al., 2011).

Aromatic compounds (benzene 40 %, ethylbenzene 29 %, C9-aromatics 24 % and xylenes 20 % on average) are also well represented by this factor. While they enter in the composition of fossil fuel combustion by constituting the unburned fraction of vehicle exhaust emissions as for C5-alkanes  (Buzcu and Fraser, 2006), their proportion in factor 4 does not compare with the one of traffic emissions derived from canister measurements along the Barbaros (Bd) (Figure 9). Solvent use activities from domestic or industrial sector can also emit aromatics higher than C6. Therefore, the presence of aromatics in factor 4

would be rather related to solvent use activities. Acetonitrile, highly present in factor 4, is usually used as a biomass burning tracer (Holzinger et al., 1999).

The diurnal profile is characterized by an increase in concentration during the day and a decrease in concentrations at night (Figure 8). When looking at external variables, this factor correlates well with $SO_2$ (r = 0.5), which is a tracer of industrial emissions and ship emissions (Lee et al., 2011) but neither with $NO_x$ (r = 0.25) nor CO (r = -0.06). Indeed the city of Istanbul

experiences the highest industrial activities in the country (Markakis et al., 2012) while the Bosphorus strait is 500 m away. The diurnal shape is also similar to the one of ship traffic counts (Figure S7 of the supplement material) which follows the one of road traffic.

The potential influence of ship emissions has been investigated by looking at the ratio of V/Ni derived from the elemental composition of $PM_{2.5}$. This ratio have been commonly used as a tracer of ship emissions influence (Viana et al., 2014; Pey et

al., 2013; Becagli et al., 2012). While some papers assume that a ratio of 3 usually signs the impact of ship emissions (Mazzei et al., 2008; Pandolfi et al., 2011) , a deeper analysis of the literature suggests that this ratio is highly variable from 0.7 up to 4.5 (Isakson et al., 2001; Agrawal et al.,2008) When plotting particulate V versus Ni concentrations integrated over a 24h-period in September 2014, a ratio of 2.72 ± 0.89 is found (see Figure S8 of the supplement material). However, the scatterplot of 6h-integrated data reveals a more scattered distribution of points on both sides of the fitting line. First, the derived V-to-Ni

value seems to be controlled by the sampling time resolution and a fixed value cannot be used as an evidence of ship emission influence. Second, while the Istanbul points lie between the upper and lower limits some of them are higher than 4.5. Other sources like coal combustion can affect the V and Ni distribution (Oztürk et al., in preparation). Finally, by comparing this factor (C5-C7 alkanes, aromatic compounds, 1- pentene and acetone) to the one of ship emissions at berth in Jingtang port (Xiao et al. 2018), no similarity was found. While an influence of ship emissions is not excluded, there is no direct evidence

from our measurements.

The analysis of the time series shows that the background level of this factor varies as a function of the meteorological period (Figure 7). During the period 2, a strong increase in minimum concentrations is observed, which may be related to the lower wind speed that favors the stagnation of pollutants. We assume that secondary production affects this factor since the presence of many secondary compounds (MACR+MVK, MEK, acetaldehyde etc.) is observed. To conclude, this factor can be related

to a combination of primary and secondary anthropogenic (combustion, industrial and ship emissions) as well as biogenic





emissions. By considering the different origins of species and its diurnal emission, this factor was labelled as "mixed regional emissions factor". It represents 36.3 % of the total VOC contribution.

By increasing the number of factors, isoprene was isolated into a 7-factor solution. By increasing the number of factors to 8, isoprene was not isolated anymore (Figure S9). By using only the PTRMS data (10 min time resolution) for PMF run, isoprene
and its oxidation products have been isolated as well. Synchronizing the PTRMS data with the GC-FID time step (30 minutes resolution) degrades the time resolution and smooths the variability of the data and, consequently, the ability of the PMF model to isolate biogenic emissions from other sources.

**Factor 5: Road Transport**

The profile of factor 5 exhibits high contributions of pentanes (iso, n and 2-methyl) with on average 84 % of their variability
explained by this factor (Figure 6), followed by n-heptane (58 %). The 32 % of the variability of 1-pentene is also explained by this factor. Aromatic compounds, such as ethylbenzene (45 %), o-xylenes, (m+p)-xylenes (47 %), C9-aromatics (37 %), benzene (28 %) and toluene (28 %), which are considered as typical fossil fuel combustion products (Sigsby et al., 1987) are also predominant species in this factor. Isopentane is one of the most abundant VOC in the traffic related sources (Buzcu and Fraser, 2006).

To help in identifying the main sources related to this factor, a comparison between its profile and the one obtained from near-source traffic measurements was performed (see Figure 7). Contrary to factor 4 (mixed regional emissions), both profiles are similar. Factor 5 is much more enriched in aromatics compared to C5-alkanes by almost a factor of 2.

The diurnal profile of factor 5 showed an increase in concentrations from midnight until sunrise and an almost constant concentration during the rest of the day with several small peaks (Figure 8). Morning peaks (6:30 and 9:30) and a night peak
(19:30) are observed. This increase in concentrations corresponds to the morning and evening traffic rush hour periods. After 18:30 the absolute concentration of this factor stayed high and increased for several reasons: ongoing emissions until 3h30, lower photochemical reactions and atmospheric dynamics (the swallower boundary layer leads to more accumulation of pollutants at night. Lower concentrations are observed during late morning until 18:30. The reduction of concentration of this factor during the day could be explained by dilution process and OH oxidation process.

The time series shows a period of peak and a relatively high contribution of this factor in the night of 09/26 to 09/27 (Figure 7). The factor 5 is the closest to traffic related source which covers exhaust emissions and gasoline evaporation emissions. However, the contribution of this factor does not correlate with the traffic tracer $NO_x$. This factor represents 15.8 % of the total VOC contribution.

As it was discussed in Yuan et al., (2012), the effect of photochemistry on factors composition had been analyzed.
Nevertheless, no clear evidence from photochemistry was founded on the Istanbul PMF factor's contributions.



### 3.3.2 Sensitivity tests to evaluate PMF results

The sensitivity of PMF results on input data have been tested in order to evaluate the representativeness of the PMF results. These tests evaluates the effect of some time period selection (periods 1, 2 and 3, nighttime peaks), the incorporation of species with a high number of missing data and uncertainties.

Table 2 summarizes the contribution of each factor for every sensitivity scenario as well as the values of the correlations between observed vs modelled concentrations.

The reference run has the best fit with $r^2_{total} = 0.97$ and more than 82 % of species were well reconstructed by the PMF ($r^2 \geq 0.5$). While the sensitivity tests are statistically less performant than the reference run, they all show that the same factors are extracted and identified even though the relative contributions could be slightly modified by ±15%.

Since benzene is a tracer of combustion, scatterplots of representative VOC versus benzene are reported in Figure 10. Benzene has a good correlation with pentanes and aromatic compounds ($0.59 \leq r^2 \leq 0.73$).

Figure 10 displays scatterplots of selected VOC vs benzene. Ratios of the selected VOC over benzene within the associated PMF factors are also reported.

All selected scatterplots are delimited by VOC/benzene ratios derived from PMF factors which means that these species are
emitted by a combination of sources.

### 3.3 Determination of emission ratios and their spatial variability

The determination of emission ratios (ER) is a useful constraint to evaluate emission inventories (Warneke et al., 2007; Borbon et al., 2013). The emission ratio is the ratio of a selected VOC with a reference compound that does not undergo photochemical processing mostly CO or acetylene due to their low reactivity at urban scale and as tracers of incomplete combustion (Borbon
et al., 2013; Salameh et al., 2017). The linear regression fit method (LRF) is a commonly used method to calculate emission ratios: the ER corresponds to the slope of the scatter plot between a given VOC vs CO or acetylene (Borbon et al., 2013; Salameh et al., 2017). Another method is to use the photochemical age method (de Gouw, 2005; de Gouw et al., 2018; Warneke et al., 2007; Borbon et al., 2013). In this study, poor correlation between targets VOC and CO is found ($r^2 \leq 0.16$) as could be deduced from the time series analysis (see section 3.2.2). As a consequence the LRF method cannot be applied. Here, the
median value of each VOC-to-CO ratios during all the observation period at Besiktas have been used. The background value of CO's mixing ratio (minimum value of CO over the measurement period) has been removed.

The median of the ratios VOC/CO in Istanbul and the emission ratios of VOC/CO in Los Angeles, United States (de Gouw et al., 2017, 2018), in Paris, France (Borbon et al., 2013) in Mexico city, Mexico (Bon et al., 2011) and in Beirut, Lebanon (Salameh et al., 2017) are displayed in table 3.

Emission ratios in the other megacities are most of the time within the same range or several factors greater than in Istanbul.

However, emission ratios are in the same range within a factor of 2 for Istanbul and Los Angeles for all the selected VOC except for benzene, pentanes (iso/n). For Istanbul and Paris, emission ratios are in the same range within a factor of 2 for only



n-hexane and aromatics (benzene, ethylbenzene and o-xylene). For Istanbul and Mexico, emission ratios are in the same range within a factor of 2 for some aromatics (toluene and benzene), alkanes (isopentane, n-hexane and n-heptane) and 1,3-butadiene

and for Istanbul and Beirut emission ratios are in the same range within a factor of 2 for alkanes (iso/n-butane, n-pentane, n-hexane, n-heptane) and 1,3 butadiene.

### 3.4 Evaluation of emission inventories

In this section, we estimated VOC emissions from the emissions ratios of VOC and CO (calculated in section 3.5) and compared these results to three downscaled global emission inventories : EDGAR (Crippa et al.. 2018) for 2012, MACCity

(Granier et al.. 2011) for 2014 and ACCMIP (Atmospheric Chemistry and Climate Model Intercomparison Project) (Lamarque et al.. 2010) for 2000 (figure 12.a,b and c). Emission data for ACCMIP and MACCity inventories are available in the ECCAD database (http://eccad.aeris-data.fr/), and the one for EDGAR inventory is available in the EDGAR database (http://edgar.jrc.ec.europa.eu/).

The estimated emissions from the observations was calculated by using the method described in (Salameh et al.. 2016).

$$VOC_{estimated} = ratio\left(\frac{VOC}{CO}\right)_{observation} \times CO_{inventory} \qquad (4)$$

$VOC_{estimated}$ is either individual or group of VOC in tons/year.

$CO_{inventory}$ is either the CO from ACCMIP (in Tg/year), MACCity (in Tg/year), or EDGAR (in tons/year).

VOC/CO (ppb of VOC per ppb of CO) is the ratio calculated in section 3.4; for the road transport emission estimation, VOC/CO is the ratio of the sum of all the VOC's concentrations in the PMF factor and that of CO measured in this experiment.

We then access the road transport emission from PMF and compared it to ACCMIP and EDGAR global emissions (figure 11.d).

The annual VOC and CO emissions for EDGAR (0.1 x 0.1 resolution) was determined by summing the emissions of 12 grids (points of measure) encompassing the sampling site (longitude between 28.9 and 29.1°; latitude between 40.9 and 41.2°). For ACCMIP and MACCITY the emissions values for the city of Istanbul was taken as available in the ECCAD database.

Further information about the emissions inventory will be found in table 3 of the supplement material.

Figure 11 shows the comparison of the estimated emissions of some speciated VOCs derived from observations and PMF for the road transport sector vs global emissions from the inventories. The annual VOC emissions are either within the same range or overestimated for all the hydrocarbons except for C9-aromatics for EDGAR inventory (figure 11.a). The overestimation varied from 4 to 7 times (xylenes and benzene respectively) for EDGAR inventory (figure 11.a), from 2 to 26 times (butanes

and benzene respectively) for ACCMIP inventory (figure 11.c), and was more than 3 times (C ≥ 4 alkanes) for MACCity inventory (figure 11.b). On the opposite, emissions of oxygenated compounds are underestimated with a factor of 10 for ACCMIP inventory and up to 26 for MACCity inventory; it was within the same range for EDGAR inventory.



In the road transport emission evaluation, EDGAR inventory was found to be better than ACCMIP inventory with most of the compounds within the range of 3 except for benzene.

Very few VOCs were used for the emission evaluation compared to those used by the global emission inventory which can be a cause of the overestimation. However, while the same general conclusion can be given about the underestimation of the emissions calculated from our observations, global emissions are not consistent between each other. ACCMIP overestimated the pentanes emissions by a factor of 3 while the same emissions was found for EDGAR inventory. Xylenes was also overestimated by a factor of 4 and 6 for EDGAR and ACCMIP inventories respectively.

**4 Conclusion**

VOC measurements were performed in Istanbul (Turkey) at an urban site in Besiktas in September 2014. The VOC measurements instruments include an AIRMOVOC GC-FID and a PTRMS at the super site completed with sorbent tubes and canisters within the megacity close to major emission sources. 23 of the 70 NMHCs quantified were continuously collected at the urban site.

During the intense field campaign, three periods had been selected from the meteorological parameters analysis and limited by two transitional periods. VOC variabilities were driven by the meteorological conditions observed, with higher concentrations during period 2 (under continental influence) and lower concentrations during period 1 and 3 (under marine influence). Also, most of the VOC were characterized by an increased in concentrations at night and early morning and lower concentrations during the day.

The average composition of VOC is mainly composed of OVOC which represent 43.9 % of the total VOC mixing ratio observed), followed by alkanes (26.33 %), aromatic compounds (20.66 %), alkenes (4.81 %), terpenes (3.44 %) and acetonitrile (0.84 %). The average atmospheric composition of anthropogenic VOC is similar to those observed in European megacities like Paris and London, suggesting the impact of traffic emissions for those compounds. However, multiple evidences of the impact of sources other than traffic like industrial activities under continental and south-southwesterly wind regimes or ship

emissions on IVOC loads have been found. This evidence was also observed in the time series analysis where the influence of multiple sources other than traffic was also suggested.

Time series analyses suggest the influence of multiple sources other than traffic on VOC concentrations at the supersite and likely industrial and/or port activities and the fact that traffic is not the most dominant source in this study.

Five factors have been extracted by the model and then compared to source profiles established by off-line near-source

measurements. These results also confirmed that road transport is not the dominant source by only explaining 15.8 % (factor 5) of measured VOC concentrations, differing to the local emission inventory. Other factors as sources resolved by the model were Toluene (14.2 %), a Biogenic terpenes (7.8 %), a Natural gas evaporation (25.9 %, mainly composed of butanes) and a last factor characterized by Mixed regional emissions (36.3 % and composed of most of the species). Evaluating the PMF




results, there is no evidence of the impact of ship emissions on VOC distribution. It is also shown that the commonly used ship
emission tracer derived from $PM_{2.5}$ composition should be used with cautious.

Several sensitivity tests on PMF results based on input data have been carried out to evaluate the effect of the time resolution
when combining different instrumentation measurements. Sensitivity tests also analyzed the impacts due to the number of
missing values and the number of species integrated in each model run. While some sensitivity tests are less performant than
the reference run, they all showed that the same factors are identified even though their relative contributions could be slightly
modified.

Considering our results and knowing that the measurement period was quite short, long-term measurements at different time
of the year will be valuable to assess the seasonality effect on source contributions in Istanbul.

Correlations of selected VOC vs benzene showed that the scatter plots of selected VOC vs benzene were delimited by PMF
factors suggesting that these compounds come from different emission sources.

Emission ratios (ER) of VOC relative to carbon monoxide (CO) were obtained and compared to that of many megacities.
These ratios in the other megacities are most of the time within the same range or several factors greater than in Istanbul. Since
some inconsistencies related to the main VOC sources in Istanbul were found, the emission ratios obtained here were used to
estimate VOC emissions by using EDGAR, ACCMIP and MACCity inventories. The results showed that the estimated annual
VOC emissions were underestimated for oxygenated compounds and within the same range or overestimated for all the
hydrocarbons except for C9 aromatics in EDGAR inventory. Inconsistency were also found among the different global
emissions. EDGAR inventory was found to be more accurate than ACCMIP's for the road transport factor emission's
evaluation.

**Author contribution**

Baye Thera is a PhD student; she is responsible for the data analysis presented work, Pamela Dominutti supported the data
analysis, Fatma Öztürk was the Turkish PI of the TRANSEMED-Istanbul campaign. She dealt with the implementation of the
super site and the aerosol measurements, Thérèse Salameh supported the PMF 670 output analysis, Stéphane Sauvage was one
of the French partner of the project ; he took part to the field campaign and supervised the analysis of VOC samples at the
laboratory, Charbel Afif was the Lebanese partner of the project ; he took part of the field campaign for trace gas analysis and
to the scientific discussions, Banu Çetin is one of the Turkish partner ; together with Fatma Ozturk she was in charge of the
supersite implementation and took part to the field campaign, Cécile Gaimoz was in charge of the trace gas measurements 675
(GCFID, PTRMS) on the field and chromatogram's treatment, Melek Keleş is a PhD student ; she took in charge the PM
measurements on the field and their analysis at the laboratory, Stéphanie Evan implemented the FLEXPART model and Agnès
Borbon is the PI of the TRANSEMED project. She coordinates the project and supervises Baye Thera's PhD.


**Acknowledgement:** The field campaign in Istanbul was supported by the ENVIMED/MISTRALS 2013 call. This work is also part of the ChArMEx
program. ChArMEx is the atmospheric component of the French multidisciplinary program MISTRALS (Mediterranean Integrated Studies at Regional and



Local Scales). ChArMEx-France was principally funded by INSU, ADEME, ANR, CNES, CTC (Corsica region), EU/FEDER, Météo-France, and CEA. This study is also partly supported by the Scientific and Technological Research Council of Turkey (TUBITAK) with the project number of 113Y025. Moreover,

the grant provided by Bolu Abant Izzet Baysal University (BAIBU) Scientific Research Projects Coordination Office (BAP) are also acknowledged for grant number of 2015.09.02.825.

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



Table 1: Statistical summary of VOC concentrations (in ppbv) measured at the urban site of Besiktas from September 17[th] to September 30[th]. The initials N, GC, P, T and C stands for the number of samples and instruments measured by the GCFID, PTRMS, tubes and canisters respectively.

| | Species | Min | 25th percentile | Median | Mean | 75th percentile | Max | σ | N/instrument |
|---|---|---|---|---|---|---|---|---|---|
| Alkanes | isobutane | 0.11 | 0.47 | 0.69 | 0.96 | 1.04 | 7.68 | 0.88 | 549/GC |
| | n-butane | 0.34 | 1.05 | 1.48 | 2.00 | 2.23 | 12.2 | 1.67 | 551/ GC |
| | isopentane | 0.14 | 0.55 | 0.82 | 1.41 | 1.36 | 19.3 | 2.14 | 546/ GC |
| | n-pentane | 0.07 | 0.20 | 0.30 | 0.55 | 0.56 | 7.50 | 0.82 | 544/ GC |
| | n-hexane | 0.17 | 0.24 | 0.34 | 0.44 | 0.53 | 2.53 | 0.33 | 205/ GC |
| | n-heptane | 0.08 | 0.09 | 0.13 | 0.21 | 0.22 | 1.23 | 0.20 | 166/ GC |
| | 2-methyl-pentane | 0.06 | 0.08 | 0.11 | 0.21 | 0.17 | 6.06 | 0.58 | 410/ GC |
| Alkenes | 1,3-butadiene | 0.11 | 0.13 | 0.15 | 0.19 | 0.21 | 0.71 | 0.11 | 126/ GC |
| | 1-pentene | 0.11 | 0.13 | 0.15 | 0.19 | 0.19 | 1.19 | 0.13 | 226/ GC |
| | isoprene | 0.13 | 0.34 | 0.55 | 0.68 | 0.81 | 3.24 | 0.52 | 580/P |
| Aromatics | benzene | 0.07 | 0.18 | 0.23 | 0.31 | 0.32 | 2.54 | 0.30 | 580/P |
| | toluene | 0.16 | 0.83 | 1.25 | 2.27 | 2.38 | 23.5 | 2.92 | 580/P |
| | ethylbenzene | 0.11 | 0.14 | 0.18 | 0.26 | 0.27 | 1.68 | 0.23 | 350/GC |
| | (m+p)-xylenes | 0.20 | 0.46 | 0.63 | 0.85 | 0.93 | 6.74 | 0.77 | 530/GC |
| | o-xylene | 0.13 | 0.20 | 0.27 | 0.38 | 0.41 | 3.17 | 0.38 | 424/GC |
| | C9 aromatics (m/z121) | 0.07 | 0.24 | 0.33 | 0.45 | 0.48 | 3.31 | 0.44 | 580/P |
| Oxygenated compounds | methanol (m/z 33) | 1.20 | 3.49 | 4.31 | 4.92 | 6.00 | 19.3 | 2.28 | 580/P |
| | acetaldehyde (m/z 45) | 0.24 | 0.98 | 1.39 | 1.59 | 1.81 | 14.6 | 1.12 | 580/P |
| | acetone | 1.26 | 1.88 | 2.33 | 2.63 | 2.98 | 19.7 | 1.30 | 580/P |
| | MACR+MVK (m/z 73) | 0.01 | 0.07 | 0.09 | 0.12 | 0.14 | 0.67 | 0.09 | 580/P |
| | MEK (m/z 71) | 0.11 | 0.22 | 0.30 | 0.38 | 0.43 | 2.27 | 0.28 | 580/P |
| Nitrile | Acetonitrile (m/z 42) | 0.09 | 0.14 | 0.17 | 0.18 | 0.20 | 0.68 | 0.06 | 580/P |
| Terpenes | Terpenes (m/z 137) | 0.06 | 0.22 | 0.33 | 0.76 | 0.91 | 5.19 | 0.91 | 580/P |





Table 2: Contributions of each factor on each sensitivity test (in percentage)

| Scenario | Reference run | All data (including night of 09/24 to 0925) | Only data during Periods 1 & 3 | Only data during Period 2 | Data with missing values > 60% excluded | Data with missing values > 30% excluded |
|---|---|---|---|---|---|---|
| $r^2_{total}$ obs. vs mod. | 0.97 | 0.86 | 0.97 | 0.86 | 0.90 | 0.92 |
| % of species with $r^2 \geq 0.5$ | 82.6 % | 56.5 % | 65.2 % | 60.9 % | 65.0 % | 70.6 % |
| F1. Toluene | 14.2 | 14.8 | 13.4 | 8.9 | 15.4 | 16.4 |
| F2. Terpenes | 7.8 | 11.5 | 12.7 | 12.1 | 5.5 | 6.9 |
| F3. Natural gas evaporation | 25.9 | 11.4 | 26.7 | 13.6 | 25.8 | 22.3 |
| F4. Mixed regional emissions | 36.3 | 47.7 | 38.2 | 45.5 | 40.2 | 42.7 |
| F5. Road transport | 15.8 | 14.5 | 9.1 | 19.8 | 13.1 | 11.8 |
| Contributions | | | | | | |

Table 3: Urban emissions rations for VOC quantified during our field measurement campaign versus carbon monoxide compared to the values obtained during field studies in Los Angeles, Paris and Mexico from the literature. Bolded value are within the same range by a factor of ±2.

| VOC | Istanbul September 2014 median ratio (pptv VOC [ppbv CO]$^{-1}$ This study | Los Angeles May-June 2010 (de Gouw et al., 2018; Borbon et al., 2013) (ppbv[ppbvCO]$^{-1}$) | Paris July 2009 (Borbon et al.,2013) (ppbv[ppbvCO]$^{-1}$) | Mexico March 2009 (Bon et al., 2011b) (pptv[ppbvCO]$^{-1}$) | Beirut January 2012 (Salameh et al., 2017) (pptv[ppbvCO]$^{-1}$) |
|---|---|---|---|---|---|
| isobutane | **1.93** | **3.08** | 4.53 | 7.2 | **3.30** |
| n-butane | **4.25** | **4.42** | 10.1 | 21.7 | **6.70** |
| Isopentane | **2.27** | 8.69 | 10.8 | **3.3** | 5.30 |
| n-pentane | **0.86** | 3.26 | 3.08 | 2.5 | **1.10** |
| n-hexane | **0.78** | **1.13** | 1.15 | 1.49 | 0.90 |
| n-heptane | **0.32** | | 2.03 | **0.36** | **0.40** |





| 2-methyl-pentane | 0.31 | | 1.29 | 1.33 | 1.20 |
| 1,3-butadiene | **0.37** | **0.35** | **0.39** | **0.278** | 0.50 |
| 1-pentene | 0.38 | | | 0.152 | 1.40 |
| Benzene | **0.66** | **1.30** | **1.07** | **1.21** | 2.00 |
| Toluene | **3.94** | **3.18** | 12.3 | **4.2** | 11.10 |
| Ethylbenzene | **0.51** | **0.57** | **0.95** | | 1.40 |
| (m+p)- xylenes | **1.70** | **1.79** | 4.59 | | 4.80 |
| o-xylene | **0.69** | **0.67** | **1.09** | | 1.70 |
| C9-aromatics | 0.92 | | | 2.8 | |
| Methanol | **13.11** | **21.2** | | 2.1 | |
| Acetaldehyde | **3.76** | **5.42** | | 1.0 | |
| Acetone | **6.78** | **11.6** | | 0.51 | |
| MEK | **0.80** | **0.88** | | 0.29 | |



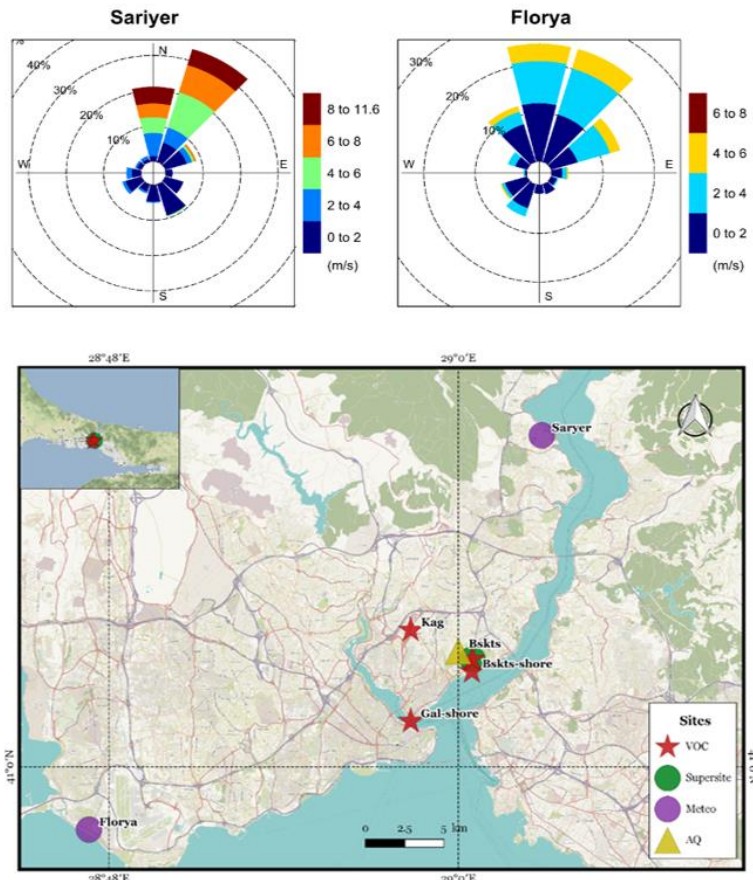


**Figure 1: Different measuring sites during the TRANSEMED campaign: Besiktas (Bskts), Kagithane (Kag), Galata (Gal), Florya (Flo) and Sariyer (Sar) and wind roses at Florya and Sariyer stations**



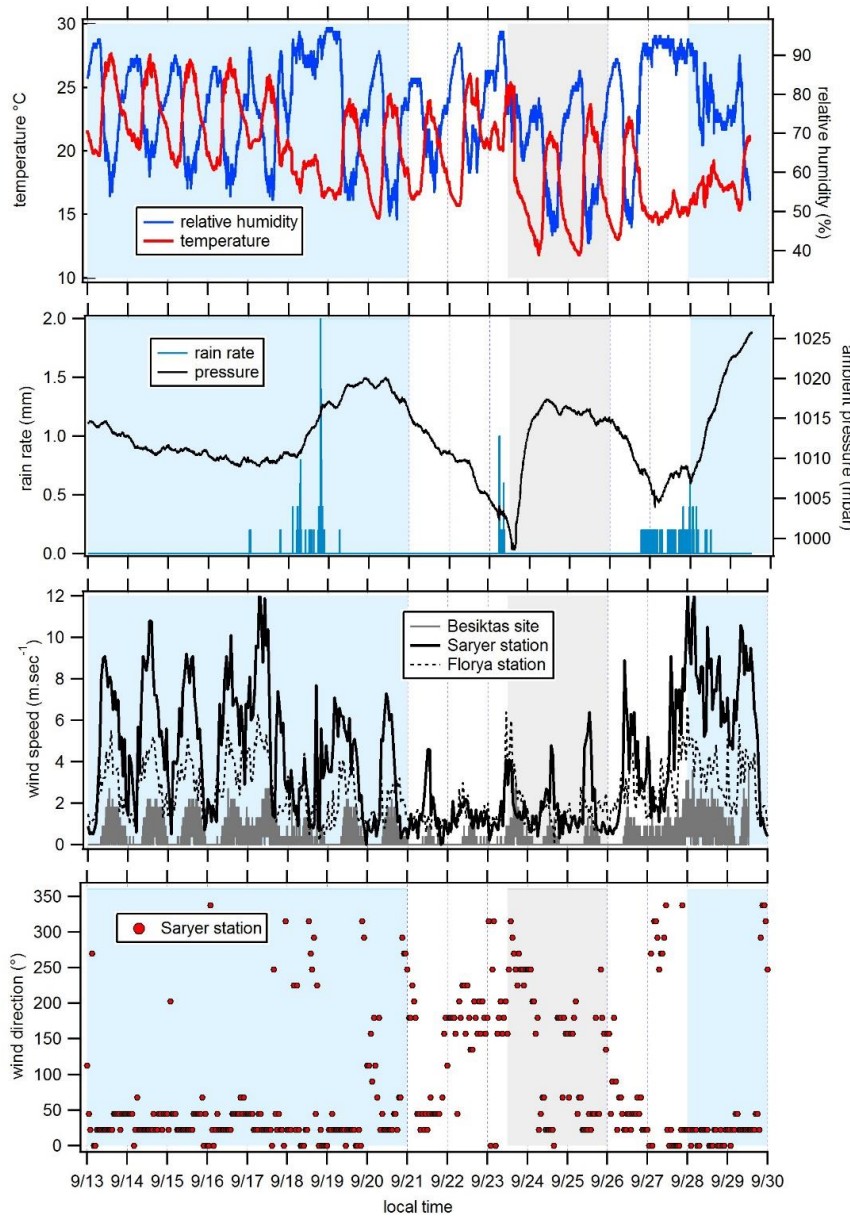

**Figure 2: Variations of meteorological parameters at the super site of Besiktas. Wind speed were recorded in the stations of Florya and Sariyer. Wind direction displayed in this Figure were recorded in the station of Sariyer. Period 1 and 3 are in blue, transition periods are in white and period 2 is in grey.**




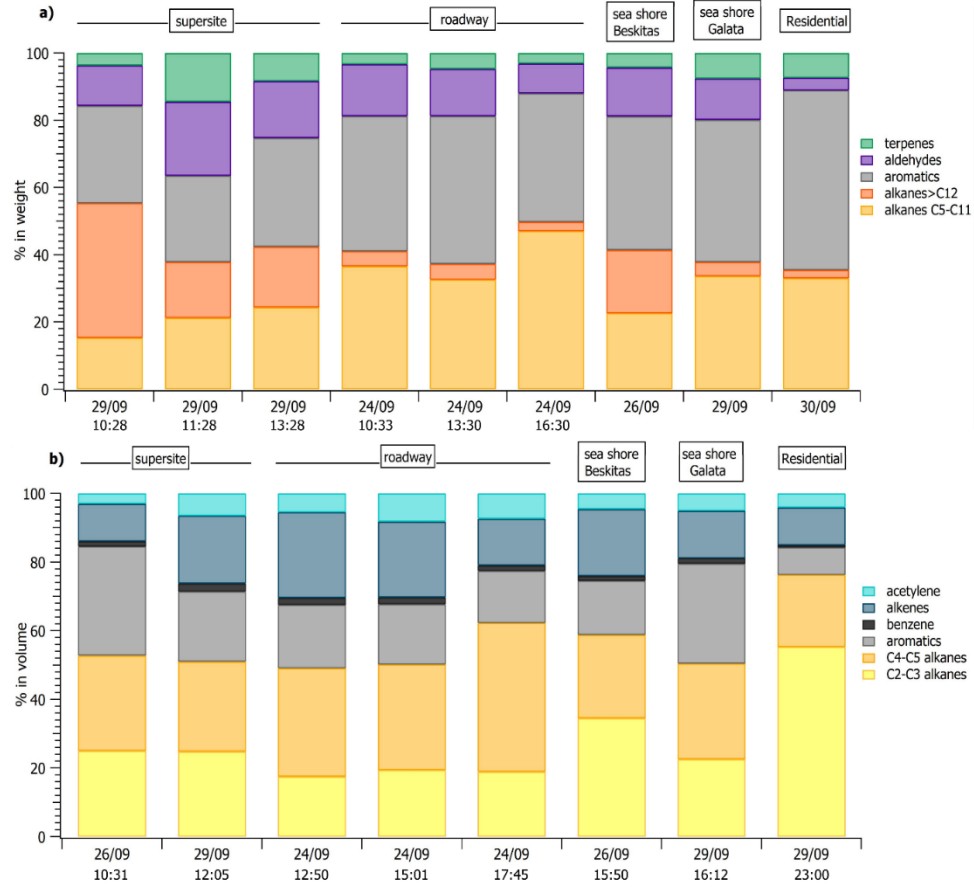

**Figure 3: VOC composition at various locations in Istanbul from tubes (a) and canister (b) sampling.**





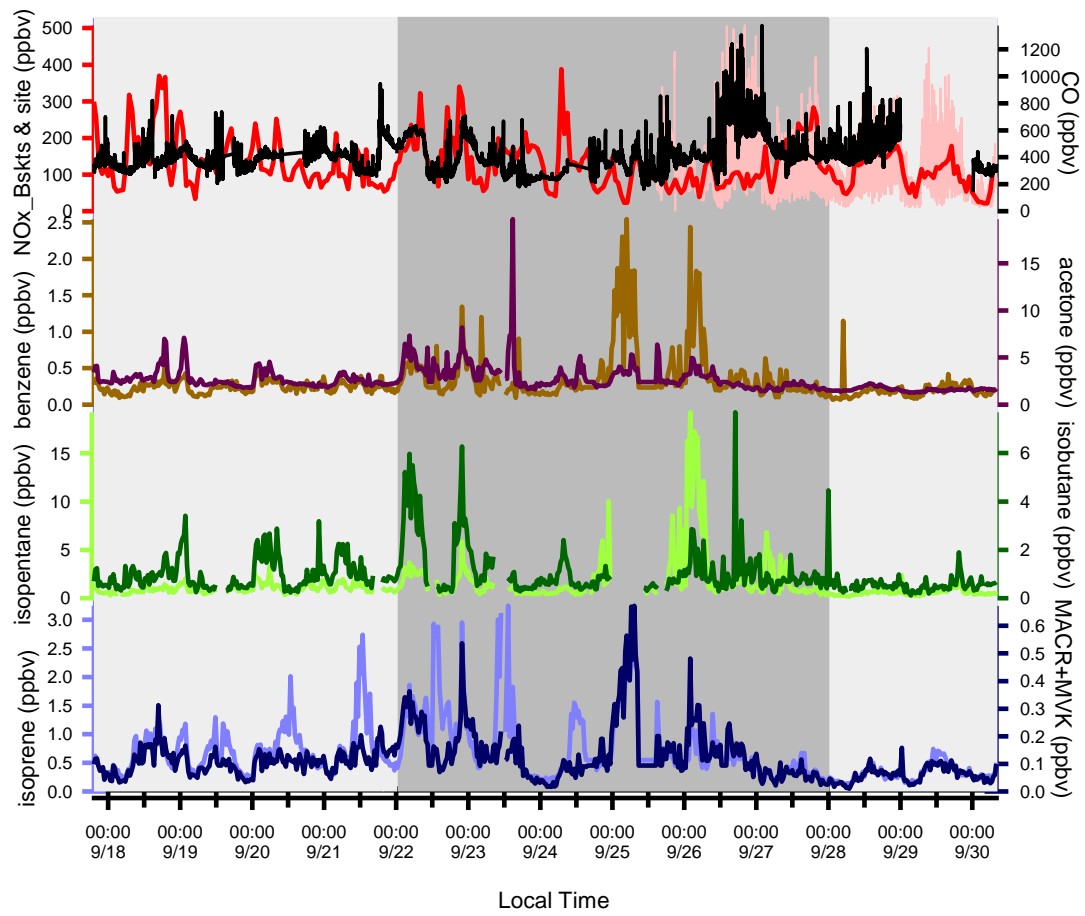

Local Time

**Figure 4: Time series of NO$_x$, CO and few VOC. Background colors represent the different periods relative to meteorological conditions (light grey period 1 and 3 and dark grey: period 2). Time series of NO$_x$ data from the super site are in pink and the ones from Besiktas site are in red.**




**Figure 5: Diurnal profiles of some selected gaseous species (ppbv). Blue shaded-areas represent the minimum and maximum diurnal concentrations over all the measurement periods, blue lines the average diurnal concentrations during period 1 and 3 and red lines the average concentrations during period 2.**



**Figure 6: Source composition profiles of the PMF factors. The concentrations (ppbv) and the percent of each species apportioned to the factor are displayed as a pale blue bar and a red color box, respectively.**





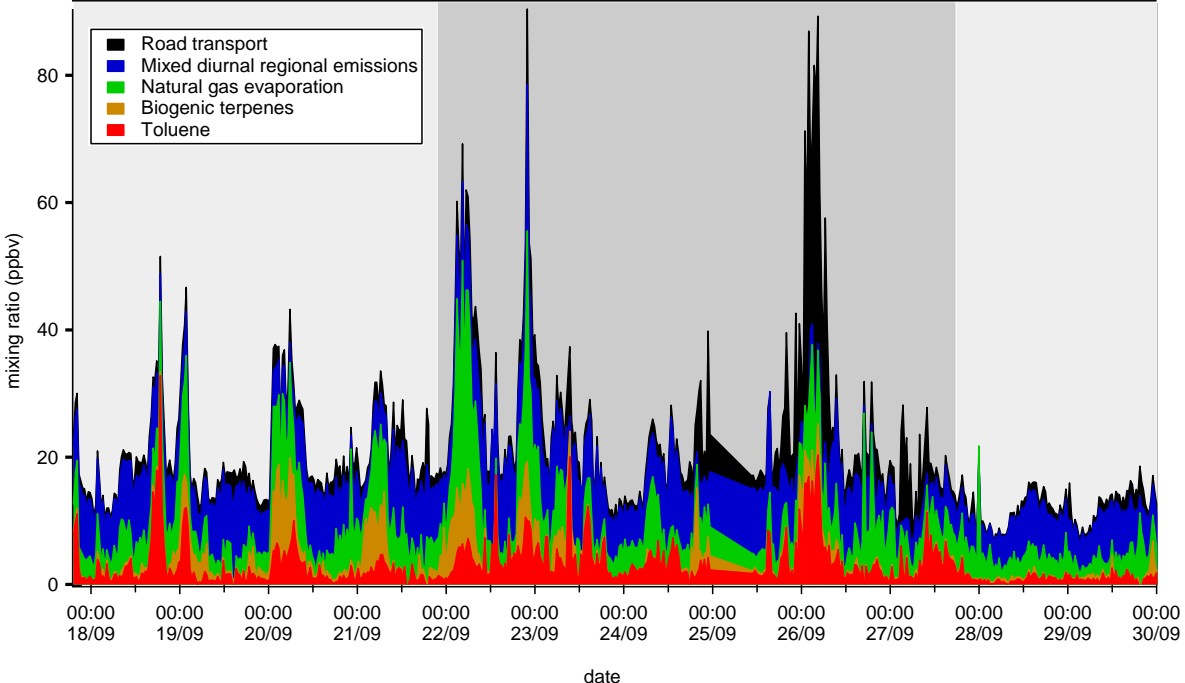

**Figure 7: Time series of factor contributions (in ppbv) extracted from the PMF.**



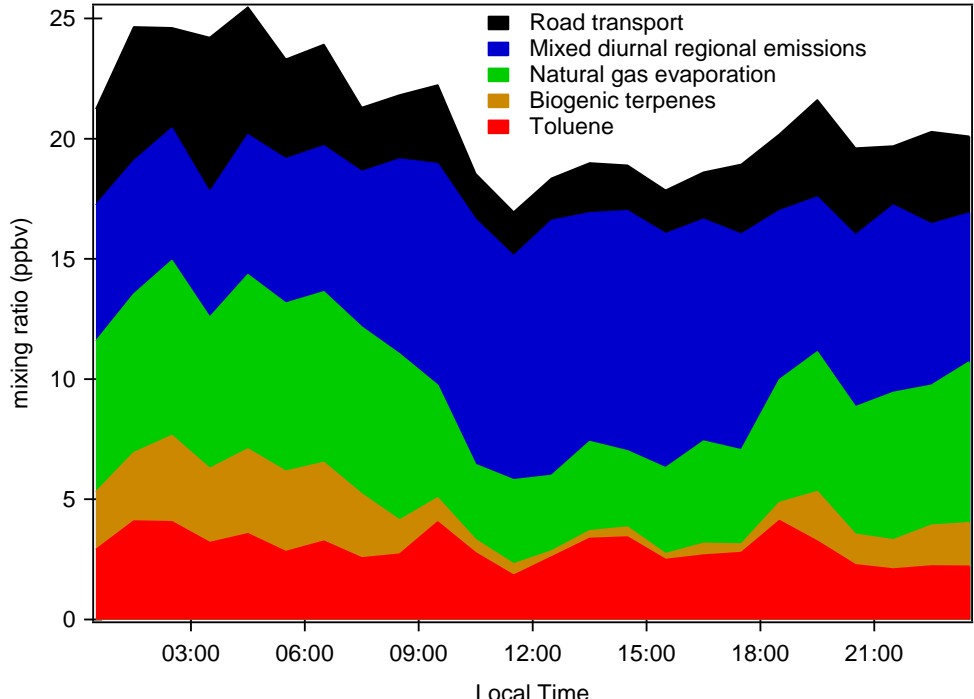

**Figure 8: Diurnal variation of source contribution (in ppbv)**






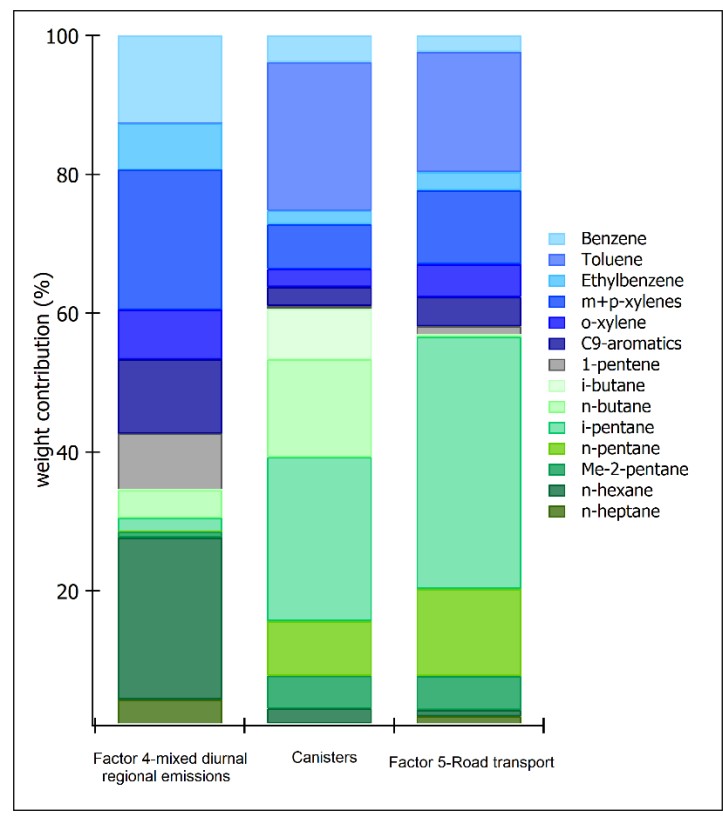

**Figure 9: Comparison of speciated profiles issued from canisters (traffic source) and Factor 4 and 5 of PMF simulations. The species contributions are expressed in percentage volume**





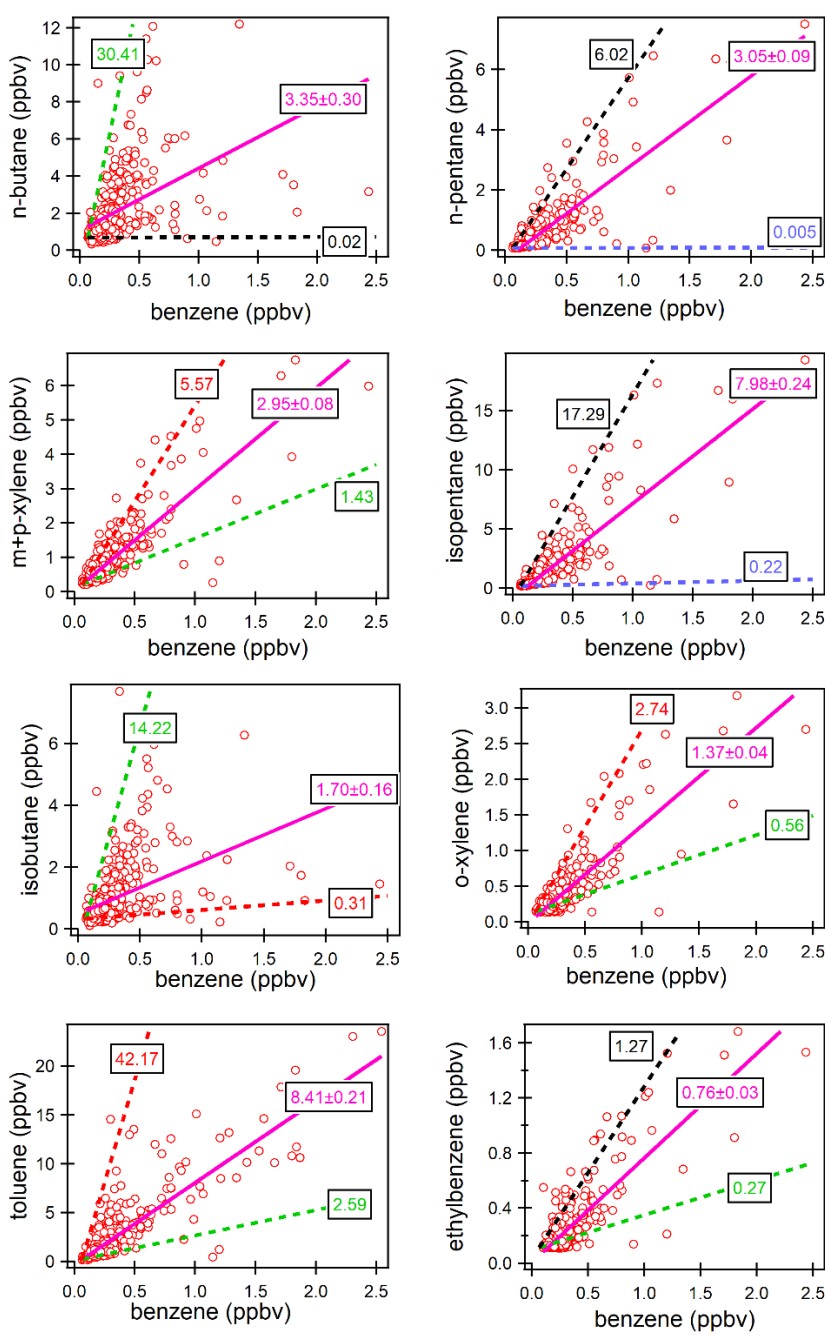


**Figure 10: Scatterplots of selected VOC versus benzene. The linear regression fit of the observations are in pink. Positive Matrix Factorization factors of this study are dotted (road transport factor is in black, mixed regional emissions in blue, natural gas evaporation in green and toluene factor in red).**





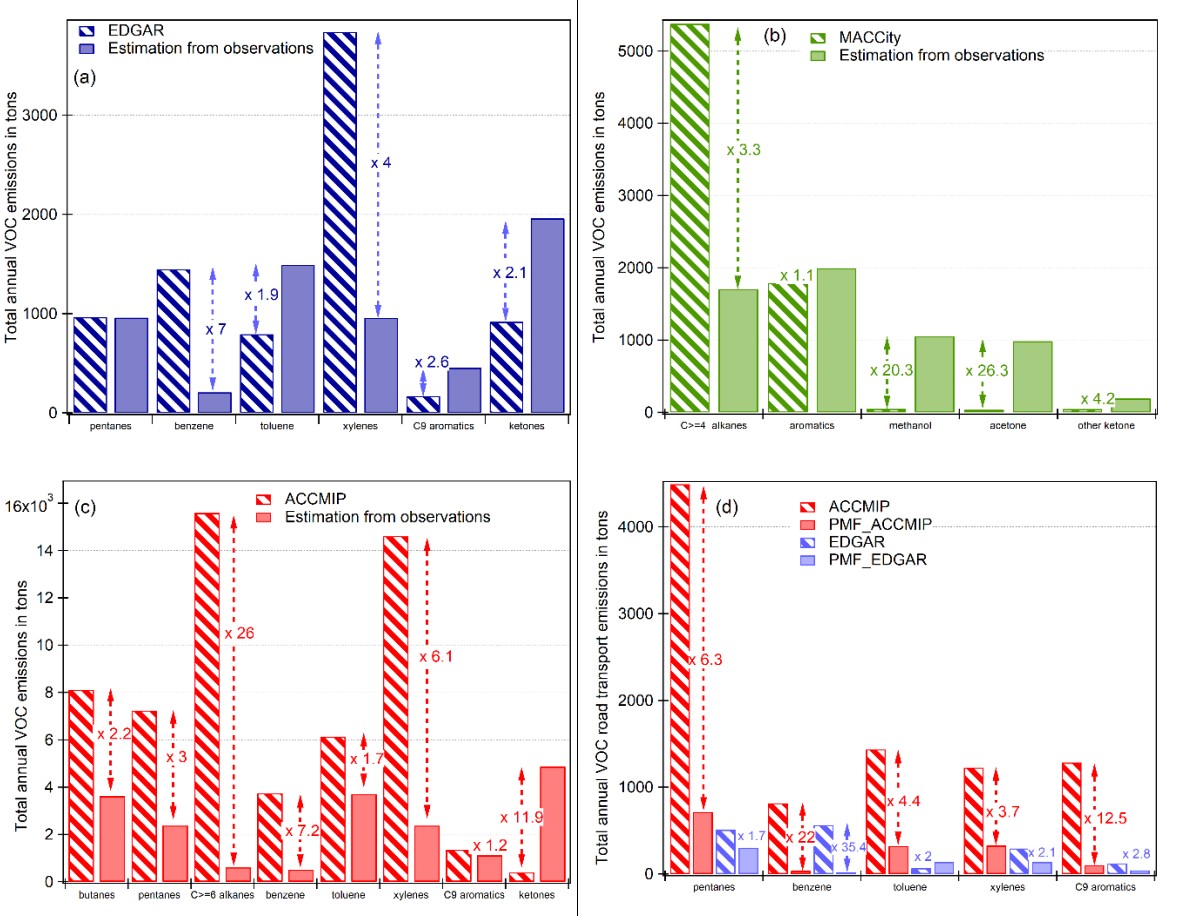

**Figure 11: Comparison of the estimated emissions inventory from observations and PMF results and global emission inventories : a) EDGAR, b) MACCity, c) ACCMIP, d) Road transport for ACCMIP and EDGAR inventory.**