# Peer review of "Composition and variability of gaseous organic pollution in the port megacity of Istanbul: source attribution, emission ratios and inventory evaluation"

_Atmospheric Chemistry and Physics, 2019_

## Referee Comment (RC1) · Anonymous Referee #2 · 17 May 2019

The paper by Thera et al. presents a 2-weeks VOCs dataset obtained in Istanbul. On-line measurements have been performed at a main site (in the city centre) and has been completed by off-line measurements. Some additional off-line samples have been taken at other sites, in order to document chemical signatures from specific sources. In a first part, the daily and day-to-day variability has been examined, based on air mass back trajectories and meteorological parameters. In a second part, a PMF analysis has been performed and the 5 determined source profiles have been presented and discussed. Finally, a comparison with the emission inventory has been

provided based on calculated emissions ratios VOCs/CO. Although quite limited in the number of measurements, the presented dataset is interesting as VOCs measurements are rather limited in this area and comparison of source contributions could bring important information to evaluate the emission inventory, which is a critical input of air quality models. Nevertheless, there is an important issue concerning the validity of this comparison, as explained below in the section "Main comment". I would recommend publication of this manuscript in ACP if this issue is solved / explained along with some other requested clarifications /improvements described in the specific comments.

Main comment

The last two sections of the papers (3.3 and 3.4) are rather short but they are potentially important as they compare the results of this study with emission inventories data. Nevertheless, currently the way the emission ratio is calculated and compared is not convincing at all. The authors say that they can not use the "linear fit regression" method in order to derive emission ratio because there is a poor correlation between targets VOC and CO. They use then the median value of each VOC to CO during all the observation period to estimate an emission ratio (before to compare it to other cities and then to emission inventories). In absence of any correlation between VOC and CO, I do not see how a ratio of median VOC/CO could be used to estimate an emission ratio...from what is representative this emission ratio? From all sources for the whole city? Indeed, as the whole dataset is used, this means that all sources are mixed; and among them traffic contributes only 15%; so how can you compare your ratio to traffic emissions from inventories? As these 2 sections are based on this emission ratio calculation, so either this one is better justified and its representativity (and limitation) is discussed, or these sections have to be removed. We note also that there is no discussion about the fact that VOCs in the inventory stand for "all VOCs" whereas only a limited number of VOC were measured..

Specific comments

[Figure]

-L64: Is the given standard deviation calculated between both calibrations or does it include the 5- ppb control points? How were the calibration coefficients applied to the data? An average value was used or an interpolated one? How was the blank value substracted? An average value was used or an interpolated one? Please clarify all these points.

-L188: I agree with the author that the variability is highly consistent for aromatics between both techniques. Nevertheless, they claim that the difference in concentrations do not exceed 20%, although the slope for toluene is 22%. In addition, we note that for benzene, there is an underestimation of about 20% of the PTRMS compared to the GC; whereas for toluene, it is the contrary (overestimation by the PTRMS). How do you explain this feature? As ethyl benzene is known to fragment on the mass of benzene, I guess we would rather expect the contrary (i.e. an overestimation of benzene on the PTRMS). Moreover, there seems to exist an even higher difference between the sorbent tube and the GC. As the ratio toluene/benzene is later on used in the paper to comment on source origins, a more careful analysis on the uncertainty associated to this ratio, due to the differences which are pointed out by the intercomparison should be made (as the ratio could be over-estimated).

-L190: where is the graph showing the comparison for isoprene?

-L228: How was calculated the 30-min data? Did it take into account the sampling time of the GC? If not, could it have an impact on the results as there was a high degree of variability of the compounds?

-L244 : With missing values higher than 40% and the use of median values instead of missing data, one can wonder about the meaningfulness of using such compounds? The authors could refer to their sensitivity tests to justify this point

-L251: Even if all details are given in the SM, please give in the main text the values used as input for uncertainties (at least the range)

-L265 to 269 : I would suggest to move the part in the methodology section

-Fig. S4 could be in the main text as it is discussed in details here

-L301: "terpens": does it include isoprene?

-L307: Give the references associate to the measurements in Paris, London and Beirut

-L308: one general comment which could be made here is that despite different years and seasons, Istanbul is quite similar to other cities, except for toluene and xylens (and this could be later on reminded when analysing the sources to discuss which of the source(s) would explain these high values in Istanbul)

-L312: what element suggests the traffic influence? ("This would suggest. . .")

-L314/Table S6: Why presenting a table of mean concentration which have been measured in different sites (and date/time). It would be more interesting to present a value (or a mean +/-std value) for a given time for each site for some compounds, this would allow a comparison with the main site.

-L331 : The section 3.2.2. could be re-arranged, in order to directly introduce the discussion on diurnal variations. In the current version, the overall variability is discussed and then the diurnal variation is discussed but this leads to some confusions (for example, L338 diurnal cycles of NOx and CO are discussed, although the figures of the diurnal cycles of are not yet properly introduced) and several repetitions (for example, the vegetation type in Istanbul. . .).

-L338 and the corresponding paragraph: The discussion of this section is not clear and might be improved, once the discussion includes as well the diurnal cycles (see previous comment). In addition the discussion focusses mainly on local meteorological conditions (wind, dispersion..) but no discussion is made on the possible influence of long-range transport. If not discussed at all, why studying Flexpart back-trajectories over such long periods?

-L338: At midday, it is not a maximum. In addition, why a midday concentrations max is expected from traffic-related compounds? Usually, a morning and an evening peak are observed

-L341: Isoprene and its oxidation products co-variate most of the time. This is not true for period 2. Be more precise in your analysis and description.

-L384 and Figure 5 : there is a large peak of benzene, isopentane, isobutene, m71 during the night of event 2. How do you interpret it? Is it due to a single event or it was observed several times? It could be useful to show toluene on this figure (directly near to benzene)

-L406 to L418: I would suggest to move this part in the methodology section

-L422 : why naming a source after a compound and not only "solvent use"?

-L422: The recent study about VOCs from petrochemical sources in urban areas (Mac Donald et al;, Science, 2018) must be referenced somewhere when discussing about solvent use

-L432: The sentence "low T/B ratio indicates the influence of traffic emissions on measured VOCs….." could be mis-leading and should be checked /re-formulated (see for example Gaeggeler et al., 2008 which says the opposite: "Another indicator for traffic emissions is a low benzene/toluene ratio (Stemmler et al., 2002)". In addition, the uncertainty of the T/B ratio should be reminded here (see comment L188). Therefore, this section should be either removed or discussed more thoroughly.

-L477: could this factor represents the "regional background"? If so, the discussion could be shortened, as there is no specific source associated and therefore no need to detail all biogenic/anthropogenic, primary/secondary source. That would avoid some vague statement. For example, L486 "these species are formed by the oxidation of primary biogenic hydrocarbons. However these oxygenated can have also primary both anthropogenic and biogenic sources". And the mention of 1,3-butadiene and 1-

pentene being emitted by plants is not so convincing in such a highly populated city.

-L480: the sensitivity study should be mentioned here (otherwise the 70% missing value would lead to the comment that this compound should not be taken into account).

-L517: it is difficult to see on the figure that a strong increase in minimum concentrations is observed during period 2

-L549: This sentence is too vague; how has it been analysed? Either remove or give a bit more information on this point

-L551: This section on sensitivity tests is important and is convincing to show that the most appropriate run has been selected. As these results are needed before, I'm wondering if it would not be more appropriate to move it at the beginning of the PMF results section (or even in the methodology part). The second part of the section (starting from L560) does not really belong to a section called "sensitivity tests" and it is not clear what it brings to the discussion. Therefore, it is suggested either to remove it or to discuss it in more details (probably in another section then).

-L551: Before to start a new section, it would be useful to have a section which comments the PMF results as a whole (for example, the contribution of the different sources compared to the other cities where levels and variability were compared...)

Technical corrections

-L53 the sentence is incomplete

-L65: "include" instead of 'includes"

-L68: rephrase (the paper by Panopoulou does not present a receptor oriented approach")

-L103: correct the English of the sentence

-L111: precise that it is at one season

-L211: There is no PM on figure S1

-L238: There is a "%" alone (with no number)

-L270/L271: Once it is referred to three main periods,, another time to four types of periods, try to be consistent in the naming and numbering

-L278 "wind" instead of "win"

-L299: some oxygenated "compounds"

-L420: "in" figure 6

-L430: there is on "(" too much

Table 1: m/z from acetone should be given as well

Figure 1 : it would be useful to have a figure zoomed on the main sites and the 2 side sites , if possible with wind roses, allowing to locate all local sources

Figure 2 and 3: it would help to harmonize the site locations names between both figures

---

## Referee Comment (RC2) · Anonymous Referee #1 · 24 May 2019

This paper investigated the gas phase pollution in an overwhelmed megacity Instabul. By performing PMF analysis they try to investigate the origin of the organic gas phase pollutants and estimate their contribution. The type of analysis itself is not unique and has been applied to various data set. However, the results are unique in a sense that they provide for the first time online gas phase analysis in Instabul, which is considered a polluted city. I recommend publication of the manuscript with a major revision concerning the PMF analysis and some other minor clarifications.

According to the times series and diurnal figures (Fig 7 and 8) all the extracted factors

have more or less the same trend: they all increase after midnight until morning and then they all decrease with almost flat behavior during the rest of the day. This is not what is expected from a PMF analysis. It seems that the separation of the different sources is poor. If all the sources are always reaching the site all together at the same time from the same direction, then PMF is unable to separate them. What is the R2 of the time series of the 5 factors between each other? How do the solutions do like in the case of 3 and 4 factors? I', afraid that if the interpolation was done in periods with a lot of missing points in a row then the PMF results may be significantly altered. How do the solutions look like if you only use the real measurements without any interpolation?

General comments:

1. Abstract: The abstract is too long. It should be shorter and more condense in a way that the reader gets only the important information. It should be more educational and provide the translation of the results.

2. Introduction: The authors use quite old literature (15-20 years old). They should enrich/replace/add more recent citations. In addition in lines 84-92, where the authors describe other VOCs studies in cities in the eastern Mediterranean, and in lines 116-118, where the authors refer to previous VOC PMF analysis, they have ignored an important study in Athens and Patras (Greece) by Kaltsonoudis et al. (2016): Temporal variability and sources of VOCs in urban areas of the eastern Mediterranean (ACP), where online VOCs were measured and PMF analysis was performed following a very alike concept with the present paper. The author should provide a comparison with respect to the results of Kaltsonoudis et al. (2016) as Athens is one of the important Mediterranean cities.

Specific comments:

3. Lines 152-153: Why did the authors use Teflon tubing instead of silcosteel or stainless steel tubing for VOC sampling? Teflon has a memory effect which could affect the measurements. What were the losses of certain VOCs in this 3m Teflon line?

4. Lines 234-235: Which data set of isoprene, benzene, toluene and C8 aromatics concentration were used in the PMF? Those taken by the PTRMS or those by GC-FID or was the average of these 2 instruments? Please explain.

5. Line 241: Linear interpolation is accepted if there is one or two missing points between two measurements. If the missing points correspond to several hours between two measurements, then the interpolation does not necessary represents the real ambient concentrations. In this meantime the concentration could have changed a lot and an interpolation could lead in fake results. So, the criterium of using or not interpolation is not the total missing points (in your case 40%) but where there points are located/ distributed between the measured points (how long a missing a period 1 hour? 5 hours? 10 hours? Please clarify that.

6. Lines 321- 330: This part is not clear to me. What is the "one VOC fingerprint" and the "other VOC fingerprint"? What is the goal of this paragraph?

7. Lines 331-387 (Section 3.2.2): This section is not well organized. For example, the authors discuss the diurnal profiles of NOx in the lines 338-339 and they go back again to NOx diurnal cycle in lines 371-375. The CO and VOC diurnal patterns are also repeated. Please first discuss the time series and then the diurnal profiles.

8. There also some contradictions. In the lines 338-339 it is written that the NOx shows a clear diurnal profile with a maxima at midday, which is wrong according to Figure 5a where the NOx profile has 2 maxima coinciding with the morning and the evening traffic. Then at lines 371-375 it is stated that the NOx profile has 2 peaks and in the evening, which is actually what is shown in Figure 5a. Please delete the wrong description.

9. Lines 334-335: NOx and CO are described as air quality trace gases? Why? So if NOx and CO are in low concentrations it means that the air quality is good enough?

10. Lines 391-398: Isoprene is reducing after 13:00-14:0, which implies possible consumption thus the corresponding isoprene products (MACR+MVK) should increase. But they don't. Please explain. Also explain why in period 2 MACR+MVK are increasing during the night. MACR+MVK have very similar profile to benzene and isopentane for both periods 1 and 2. Is it possible that m/z 71 (related to MACR+MVK) has interferences from other compounds related to anthropogenic activities? Please discuss.

11. Lines 394-395: If furans contribute to isoprene signal (m/z 69) then this is an interference of another/different compound to this m/z. It is not an anthropogenic origin of isoprene. Please correct the corresponding sentence.

12. Lines 400-402: What does it mean a secondary source? Maybe you want to replace it with origin? Please rephrase.

13. Lines 444-445: It is strange that isoprene has only 5% to the biogenic factor. This indicates that most of the signal in this m/z is probably attributed to other compounds rather than isoprene. Please discuss.

14. Lines 481. Again, are you sure it is isoprene?

15. Lines 486-487: Could you give some examples of "primary biogenic hydrocarbons"?

16. Line 497: No, the diurnal profile of the Factor 4 has the opposite behavior according to Fig 8. Please correct the text.

Technical comments:

17. Figure 2: The 2 lower graphs need a black line (y axes) on the right in order to be like the 2 upper ones.

18. Figure 3: Please increase the fond of the letters.

19: Figure 4: Please put a black line on the top of the graph all make all axes line with the same thickness. "NOx_Bskts & site" is not an appropriate name for an axis. Please replace it with "NOx".

20. Figure 5: Please replace "NOx_Bskts" and "CO_site" with "NOx" and "CO" correspondingly.

21. Figure 6: Replace "Conc." with "Concentration (ppbv)" on the left axes. Instead of Factor 1 ,2 etc. please write the names of the factors so that the reader doesn't go back and forth.

22. Figure 7. The graph needs a black line (y axes) on the right. Please increase the fond of the letters and replace "date" with "Date and time".

23. Figure 9. Please increase the fond of the letters.

---

## Author Comment (AC1) · 8 Aug 2019

**Manuscript No.** : acp-2019-74

**Title:** Composition and variability of gaseous organic pollution in the port megacity of Istanbul: source attribution, emission ratios and inventory evaluation.

**Special Issue***: Chemistry and AeRosols Mediterranean Experiments (ChArMEx) (ACP/AMT inter-journal SI)*

**Author(s):** Baye T.P Thera et al.

Dear editor,
We would like to thank the two reviewers for their helpful comments to improve the quality of the manuscript. We have reported below all the comments and have addressed them one by one. Our responses appear in blue.

**Reviewer 1:**

**Comments:**
According to the times series and diurnal figures (Fig 8 and9) all the extracted factors have more or less the same trend: they all increase after midnight until morning and then they all decrease with almost flat behavior during the rest of the day. This is not what is expected from a PMF analysis. It seems that the separation of the different sources is poor. If all the sources are always reaching the site all together at the same time from the same direction, then PMF is unable to separate them. What is the $R^2$ of the time series of the 5 factors between each other? How do the solutions do like in the case of 3 and 4 factors? I'm afraid that if the interpolation was done in periods with a lot of missing points in a row then the PMF results may be significantly altered. How do the solutions look like if you only use the real measurements without any interpolation?

Thera et al.: Reviewer 1 raises four critical points which are going to be discussed below.

(1) All the PMF factors except for the "mixed diurnal regional factor" have indeed more or less the same trend. Indeed, the PMF is sensitive to the variability of the species but by looking at the individual diurnal profiles of the factors (figure 6) the separation of the different sources is not poor. The previous representation of the factor diurnal profile could bring some confusion and the new figure 9 is more explicit. Except for the mixed diurnal regional factor, the PMF was able to distinctly separate the other factors. A relevant example is the one of pentanes and butanes for which representative diurnal profiles are reported in the figure below (figure a) extracted from the Figure 6 of the paper.

[Figure]

While their diurnal profiles almost show similar diurnal variability during periods 1 and 3 and 2, the PMF was able to isolate those compounds in two different and independent factors (factor 3 and 5) with an $R^2$ lower than 0.1 as discussed in section 3.3.2 and in the section below.

(2) The best number of factor for the PMF run has been selected rigorously as described in section 2.4.3 of the paper. It is based on common statistical criteria such as Q (residual sum of squares), IM (maximum individual Column mean), IS (maximum individual column standard deviation) as defined by Lee and al. (1999) and $R^2$ (indicator of the degree of correlation between predicted and observed concentrations). Q, IM, IS and $R^2$ were then plotted against the number of factors (from 2 to 12) in order to extract the optimal numbers of factors. Moreover we made sure that the factors were not dependent between each other. In the table below, we have reported the $R^2$ of the time series of the chosen 5 factors between each other. $R^2$ does not exceed 0.28. There is therefore no significant correlation between the factors which means that the factors are independent. A discussion on $R^2$ values between the five factors have been added in the paper in lines 267-269.

|  | Toluene | Biogenic terpenes | Natural gas evaporation | Mixed diurnal regional emissions | Road transport |
|---|---|---|---|---|---|
| Toluene | x | 0.0590 | 0.1050 | 0.0015 | 0.2630 |
| Biogenic terpenes | x | x | 0.2822 | 0.0265 | 0.0185 |
| Natural gas evaporation | x | x | x | 0,0003 | 0.0499 |
| Mixed diurnal regional emissions | x | x | x | x | 0.0650 |
| Road transport | x | x | x | x | x |

(3) The solution with 3 and 4 factors is discussed. The solution of 3 factors does not enable to separate properly the species (see figure below): factor 1 is composed mainly of pentanes and aromatics compounds, factor 2 of butanes and some aromatic compounds and factor 3 of a mixed of all compounds except for butanes and pentanes. Terpenes was distributed between all the three factors. ). The total correlation between reconstructed and measured VOC is poor for a PMF run for a solution of 3 factors ($R^2$=0.78) and species like butanes are poorly reconstructed ($R^2$= 0.1).

3 factor solution                                    4 factor solution

[Figure]

In the case of 4 factors; we have the same factors as for the solution with 5 factors except for toluene (see figure below): mixed diurnal regional factors (factor 1), terpenes (factor 2), road transport (factor 3) and natural gas evaporation (factor 4). However, Toluene is better reconstructed by the PMF with the 5 factors solution ($R^2 = 0.95$) than the 4-factors solution ($R^2 = 0.74$). Furthermore, the sensitivity tests, the PMF output uncertainties methods, and the f-peak enable us to choose the 5 factor as the optimal solution.

(4) There was one period during which there were no measurements by the GC-FID: from 09/24 at 23:48 to 09/25 at 10:18.This period corresponds to 10h and 30 minutes of missing points. However the PMF in this experiment designed as the reference run was carried out by removing this period as discussed in section 2.4.4. Note that the PMF cannot be run with any missing points; As a consequence either we will interpolate or we will replace the missing data by the median value which is more likely to alterate the results since the latter smoothes the variability. We run the PMF by replacing missing data by the median instead of interpolating. The same number and nature of factors have been found while some differences are found in factor's contribution like the ones for Natural Gas Evaporation (26% against 10%). . Comparison between both run is reported below.

**PMF reference run**

- natural gas evaporation
- Toluene
- Mixed emissions
- Terpenes
- Road transport

PMF reference run: 15,77% | 25,91% | 7,81% | 14,18% | 36,34%

No interpolation: 9,97% | 20,24% | 13,98% | 12,26% | 43,55%

However, the reference run is still the best solution since the $R^2_{total}$ of the observed vs modelled by the PMF is 0.97 for the PMF reference run against 0.90 for the PMF run with no interpolation. There were only 65 % of the species that were well reconstructed by the PMF ($R^2 \geq 0.5$) with the run with no interpolation against 83 % for the PMF reference run. Furthermore, butanes were poorly reconstructed by the PMF with the solution with no interpolation ($R^2 = 0.19$) while these species were well reconstructed by the PMF reference run($R^2 > 0.90$). A discussion on this test has been added in the sensitivity test section (2.4.5) and in Table 2.

**General comments**

1. Abstract: The abstract is too long. It should be shorter and more condense in a way that the reader gets only the important information. It should be more educational and provide the translation of the results.

   Thera et al.: The abstract has been shorten and condense by highlighting only important information as you suggested.

2. Introduction: The authors use quite old literature (15-20 years old). They should enrich/replace/add more recent citations. In addition in lines 84-92. where the authors describe other VOCs studies in cities in the eastern Mediterranean. and in lines 116-118. where the authors refer to previous VOC PMF analysis. they have ignored an important study in Athens and Patras (Greece) by Kaltsonoudis et al. (2016): Temporal variability and sources of VOCs in urban areas of the eastern Mediterranean (ACP). where online VOCs were measured and PMF analysis was performed following a very alike concept with the present paper. The author should provide a comparison with respect to the results of Kaltsonoudis et al. (2016) as Athens is one of the important Mediterranean cities.

   Thera et al:  The introduction has been enriched with more recent citation. The work of Kaltsonoudis et al. (2016) has been added in Lines 79 in addition to the other VOC studies made

in the cities of the eastern Mediterranean as well as in lines 112 where previous PMF studies were made**.**

**Specific comments:**

3. Lines 152-153: Why did the authors use Teflon tubing instead of silcosteel or stainless steel tubing for VOC sampling? Teflon has a memory effect which could affect the measurements. What were the losses of certain VOCs in this 3m Teflon line?

Thera et al: Silco-treated steel or heated stainless steel lines are the ones recommended for hydrocarbon sampling while Teflon-PFA (perfluoroalcoxy) is the one recommended for the sampling of oxygenated VOCs. See ACTRIS http://fp7.actris.eu/Portals/97/deliverables/PU/WP4_D4.4_M24.pdf. A compromise needed to be found for the PTR-MS which encompasses hydrocarbons like aromatics and oxygenated VOCs (OVOC) like acetone. We decided to use Teflon-PFA. The good consistency at ±20% between PTRMS, AIRMOVOC, canisters and tubes reported in Figure S2 suggests that the Teflon-PFA is well adapted.

4. Lines 234-235: Which data set of isoprene. benzene. toluene and C8 aromatics concentration were used in the PMF? Those taken by the PTRMS or those by GC-FID or was the average of these 2 instruments? Please explain.

Thera et al.: The data set of isoprene, benzene, toluene and C8-aromatics concentration used in the PMF are those taken by the PTRMS. One of the reason is that there were only two missing points with PTRMS data which is better for running the PMF model. The text have been modified for more clarity in lines 230: *[…] Alkanes and alkenes were measured by the GCFID while benzene. toluene, isoprene, C8- aromatic, carbonyls, alcohol, nitrile and terpenes were the ones measured by the PTRMS. For benzene. toluene and C8 aromatics the PTR-MS data were selected for the PMF run because of the smallest number of missing data […]*

5. Line 241: Linear interpolation is accepted if there is one or two missing points between two measurements. If the missing points correspond to several hours between two measurements, then the interpolation does not necessary represents the real ambient concentrations. In this meantime the concentration could have changed a lot and an interpolation could lead in fake results. So, the criterium of using or not interpolation is not the total missing points (in your case 40%) but where there points are located/ distributed between the measured points (how long a missing a period 1 hour? 5 hours? 10 hours? Please clarify that.

Thera et al.: This comment also refers to the first one. The longest period (10h and 30 min) with missing data occurred from the night of 09/24 to the morning of 09/25 for compounds measured by the GC-FID. The PMF reference run was performed by removing this period.

Depending on the compound, the missing point period can last up to 10 continuous hours like methyl-2-pentane and one full day for m+p-xylenes. We could have replaced the missing points either by the median or by interpolation. This is the reason why we only did interpolation with species that have less than 40 % of missing data. The missing data are homogeneously distributed between all the periods. We found it more accurate to replace the missing data by the interpolation which take into account previous concentration rather than by a median. Moreover, the test with or without interpolation show that even with using the median, the PMF is less performant (see previous discussion in the first answer (4)).

6. Lines 321- 330: This part is not clear to me. What is the "one VOC fingerprint" and the "other VOC fingerprint"? What is the goal of this paragraph?

Thera et al.: the term "fingerprint" was replaced by "composition" to make it clearer and some sentences have been modified. The objective of the comparison between the different composition is to show what type of source signature can be depicted at the Besiktas supersite (lines 325-358*):[…] Therefore, the analysis only focuses on VOC relative composition. The relative composition divided into major VOC chemical groups at each sites by sorbent tubes and canisters is reported on Figure 3a and 3b. respectively. The composition is variable across the megacity for the aliphatic fraction of high and intermediate volatility hydrocarbons (C2-C16). As expected. the composition of the 29/09 12:05 sample at the Besiktas site is like the ones derived from the nearby roadway side measurements highlighting the influence of road transport emissions at the supersite. Interestingly. the VOC composition of the three samples from sorbent tubes at the Besiktas site are different from the ones at the nearby roadway side with a higher proportion of IVOC. The other VOC composition of the 26/09 10:31 sample by canister is rather similar to the one from the seashore sample in Galata (29/09 16:12 sample). In the same way the VOC composition of the samples at the supersite derived from tubes are rather like the Besiktas seashore one ; for both of them. the proportion of IVOC is significant (from 15 % to 40 % in weight). While light VOC are expected to be of minor importance when considering ship emissions. the higher presence of heavier organics is however expected as observed for alkanes by Xiao et al. (2018) in ship exhaust at berth. The VOC composition comparison would thus suggest not only the impact of road traffic emissions on their composition but also the potential impact of local ship traffic emissions. Finally the composition at Besiktas is not affected by Residential emissions which are enriched in light C2-C3 alkanes (canisters) or aromatics (canisters) [...]*

7. Lines 331-387 (Section 3.2.2): This section is not well organized. For example. The authors discuss the diurnal profiles of NOx in the lines 338-339 and they go back again to NOx diurnal cycle in lines 371-375. The CO and VOC diurnal patterns are also repeated. Please first discuss the time series and then the diurnal profiles.

Thera et al.: This section has been reorganized as you suggested in lines 360-441: *[…]The variability of VOC concentrations is driven by several factors: emissions (anthropogenic or biogenic), photochemical reactions (especially with the OH radical during the day and ozone and nitrates at night for alkenes) and the dynamic of the atmosphere (including dilution due to the height of the boundary layer) (Filella and Peñuelas, 2006). The time series of inorganic trace gases (NOx and CO) and some VOC representing the diversity of sources and reactivity are reported in Figure 5. The meteorological periods 1, 2 and 3 described in the previous section 3.1 are also indicated. Because NOx at the super site were only measured from 09/25 to 09/30, data from the air quality station in Besiktas were used (see Figure 1). One should note that the time series of NOx at the supersite and at the Besiktas station are consistent.*

*Time series of NOx and CO show high concentrations but a different pattern regardless of the origin of air masses. While a daily cycle of NOx is depicted, CO does not show any clear pattern. The NO2/NOx ratio fluctuates between 0.34 to 0.93 with an average and median value of 0.53 and 0.55, respectively. These values are very high compared to what is usually found in the literature (Grice et al., 2009; Kousoulidou et al., 2008; Keuken et al., 2012) which are mostly low and below 0.50. However higher values of NO2/NOx ratio can be found in diesel passenger cars (Grice et al., 2009. Vestreng et al., 2009) and vans (Kousoulidou et al., 2008). This ratio would reflect the impact of the combustion of heavy fuels in the megacity. After road transport, cargo shipping is a second highest contributor to NOx levels according to the local/regional inventory (Markakis et al., 2012).*

*Anthropogenic VOC time series (benzene, isopentane and isobutane) exhibit a high frequency variability but usually show higher concentrations during the night especially during period 2. One cause are the very low wind speeds at night especially during period 2 (Figure 3), which would reinforce the accumulation of pollutants. Under marine influence (periods 1 and 3), VOC concentrations are the lowest, especially during period 3, which is characterized by rainy days (September 27th and 28th), high wind speed and colder temperatures (Figure 5). These conditions favor atmospheric dispersion. During transition periods and under continental influence (period 2), VOC concentrations exhibit a strong day-by-day variability with episodic nocturnal peaks especially on September 25th and 26th. While these peaks are not always concomitant between VOC and are not associated with any increase in NOx and CO levels, they occur under south and southwestern wind regimes which are unusual wind regimes according to Figure 1. This points out the potential influence of industrial and port activities other than fossil fuel combustion. For instance, maximum concentrations of butanes occurred during the period of the marine-continental regime shift with well-established southwestern wind regime on 09/22, 09/23 and on 09/26 at the end of the day. Maximum concentrations of pentanes occurred during the night of 09/26 to 09/27 like for aromatics (e.g. benzene) (Figure 5).*

*Except during transition periods, the background levels of measured trace gases are not affected by the origin of air masses. This strongly suggests that the pollutants measured during TRANSEMED-Istanbul were from local and regional sources. Finally, time series would suggest the influence of multiple local and regional sources other than traffic on VOC concentrations, likely industrial and/or port activities, at the supersite.*

*Isoprene and its oxidation products (MACR+MVK) covariate most of the time. They usually show their typical diurnal profiles with higher concentrations during the warmest days and at midday due to biogenic emission processes. Their significant correlation with temperature (R = 0.7) implies the emission from biogenic sources. Around the Besiktas site, 49.5 % of the vegetation is occupied by hardwood and hardwood mix trees while only 6 % is occupied by softwood and hardwood mix trees. While Quercus (isoprene emitter) only occupies 7.7 % of the total vegetation coverage (personal communication from Ministry of Forestry), the presence if isoprene at the supersite is probably due to the surrounding trees.*

*Except during transition periods, the background levels of measured trace gases are not affected by the origin of air masses. The background levels stay constant under continental or marine influence and regardless of the atmospheric lifetime of the species. This strongly suggests that the pollutants measured during TRANSEMED-Istanbul were from local and regional sources. Finally, time series would suggest the influence of multiple local and regional sources other than traffic on VOC concentrations, likely industrial and/or port activities, at the supersite.*

*Taking into consideration time series variability, diurnal variations have been splitted into periods 1 and 3 and period 2 for selected VOC as well as two combustion derived trace gases (NOx and CO). Diurnal profiles of atmospheric concentrations are reported in Figure 6. Local traffic counts for road transport (personal communication from Istanbul Municipality for fall 2014) and ship (https://www.marinetraffic.com/en/ais/details/ports/724/Turkey_port:ISTANBUL) are also reported in Figure S6 in the supplement material. Maritime traffic is mostly for passenger shipping (58.02%) against 16% for cargo shipping. The diurnal profiles of ship and road traffic counts are similar.*

*Generally, concentrations during period 2 are higher than the ones during periods 1 and 3 and show different diurnal patterns for some compounds. The profile of NOx is consistent with the one of traffic counts (Figure S6 of the supplement material). NOx exhibits higher concentrations during the day and lower concentrations at night for both periods with a morning peak (7:30-8:30) and one early evening peak from 17:30 (Figure 6.a). This is typical of traffic emitted compounds with morning and evening rush-hour peaks as observed in many other urban areas like Paris,*

*France in Europe (Baudic et al., 2016) or Beirut, Lebanon in Eastern Mediterranean (Salameh et al., 2016). As already depicted in time series, CO diurnal profile is different from the one of NOx. CO concentrations show higher concentrations in the late evening and lower concentrations during the day. During the day, CO is also characterized by a double peak: one in the morning (8:30) and the other one in the middle of the day (Figure 6.b). Both NOx and CO show quite similar diurnal profile between the three periods even if morning concentrations tend to be higher.*

*VOCs show different profiles from the one of NOx. Under marine influence (periods 1 and 3), primary anthropogenic VOC (ie. benzene, alkanes and other aromatics) almost exhibit a constant profile while they show higher concentration from midnight until 10:00 AM under continental influence (period 2), For instance, benzene (Figure 6.d) and isopentane (Figure 6.f) nighttime concentrations increase by four-fold compared to the levels under marine influence. In the middle of the day, the concentration levels are the same as during periods 1 and 3. The profiles of primary anthropogenic VOCs point out the complex interaction between local and regional emissions and dynamics. Period 2 points out the influence of VOC emissions other than traffic and combustion processes (no effect on NOx and CO) at night. While the influence of traffic emissions on CO and VOC cannot be excluded; it seems that their emission level is not high enough to counteract the dispersion effect during the day unlike NOx. This will be further investigated in the PMF analysis.*

*Isoprene concentrations increase immediately at sunrise and decrease at sunset during period 1 and 3 (marine influence) which indicates its well-known biogenic origin (Figure 6.g) which is light and temperature dependent. Isoprene and MACR+MVK's concentrations increase at night during period 2 like other alkanes and aromatics, suggesting their potential anthropogenic influence.*

*Provided some interferences like furans could contribute to isoprene signal by PTRMS measurements (Yuan et al., 2017), this would suggest an anthropogenic origin for isoprene. While the signals of m/z 71 are commonly attributed to the sum of MVK and MACR which are both oxidation products of isoprene under high-NO conditions, more recent GC-PTR-MS studies identified some potential interferences for MVK and MACR measurements, including crotonaldehyde in biomass-burning emissions, C5 alkenes, and C5 or higher alkanes in urban regions (Yuan et al., 2018). Such interferences cannot be ruled out here. During periods 1 and 3, MACR+MVK concentrations follow the same general pattern as of isoprene's.*

*With relatively long atmospheric lifetime, (≈ 68 days), acetone's concentration is quite constant throughout the day within period 1, a peak in the middle of the day and lower concentrations during the night for period 2 (Figure 6.c). The peak in the middle suggests the presence of a secondary origin. Acetone can have both primary and secondary source (Goldstein and Schade, 2000; Macdonald and Fall, 1993). Methanol and MEK have the same general pattern as for acetone during both periods without the peak in the middle of the day suggesting that they might have the same emission source[…].*

8. There also some contradictions. In the lines 338-339 it is written that the NOx shows a clear diurnal profile with a maxima at midday. which is wrong according to Figure 5a where the NOx profile has 2 maxima coinciding with the morning and the evening traffic. Then at lines 371-375 it is stated that the NOx profile has 2 peaks and in the evening. which is actually what is shown in Figure 5a. Please delete the wrong description.

Thera et al.: The wrong description has been deleted and the part has been rephrase in lines 409-421: *[…]The profile of NOx is consistent with the one of traffic counts (Figure S6 of the supplement material). NOx exhibits higher concentrations during the day and lower concentrations at night for both periods with a morning peak 410 (7:30-8:30) and one early evening peak from 17:30 (Figure 6.a). This is typical of traffic emitted compounds with morning and evening rush-hour peaks […]*

9. Lines 334-335: NOx and CO are described as air quality trace gases? Why? So if NOx and CO are in low concentrations it means that the air quality is good enough?

Thera et al.: By air quality trace gases we meant inorganic trace gases. We replaced air quality by inorganic trace gas in lines 363: *[…]The time series of inorganic trace gases (NOx and CO) and some VOC […]*

10. Lines 391-398: Isoprene is reducing after 13:00-14:0 which implies possible consumption thus the corresponding isoprene products (MACR+MVK) should increase. But they don't. Please explain. Also explain why in period 2 MACR+MVK are increasing during the night. MACR+MVK have very similar profile to benzene and isopentane for both periods 1 and 2. Is it possible that m/z 71 (related to MACR+MVK) has interferences from other compounds related to anthropogenic activities? Please discuss.

Thera et al.: During period 2, when isoprene decreases, its oxidation products MACR+MVK do increase. During period 2 MACR+MVK concentration increases during the night like most of the anthropogenic VOCs in this study. As discussed in the main article, this is probably due to wind regimes during this period (low wind speed that favor the accumulation of pollutants). The increase in concentration of MACR+MVK also suggest an anthropogenic origin. While the signals of m/z 71 are commonly attributed to the sum of methyl vinyl ketone (MVK) and methacrolein (MACR) which are both oxidation products of isoprene under high-NO conditions, more recent GC- PTR-MS studies identified some potential interferences for MVK and MACR measurements, including crotonaldehyde in biomass-burning emissions, C5 alkenes, and C5 or higher alkanes in urban regions (Yuan et al., 2018). Such interferences cannot be ruled out here. This discussion has been added into the main paper.

11. Lines 394-395: If furans contribute to isoprene signal (m/z 69) then this is an interference of another/different compound to this m/z. It is not an anthropogenic origin of isoprene. Please correct the corresponding sentence.

Thera et al.: the sentences has been changed in lines 431: *[…]Provided some interferences like furans do not contribute to isoprene signal by PTRMS measurements (Yuan et al.. 2017). this would suggest an anthropogenic origin for isoprene. During period 1 and 3. MACR+MVK also show high concentrations at night during period 2 like other alkanes and aromatics. suggesting their potential anthropogenic influence. During periods 1 and 3 (marine influence). MACR+MVK concentrations follow the same general pattern as of isoprene's[…]*

12. Lines 400-402: What does it mean a secondary source? Maybe you want to replace it with origin? Please rephrase.

Thera et al.: By secondary source we wanted to express secondary origin. We rephrased by replacing source by origin in lines 438: *"The peak in the middle suggests the presence of a secondary origin"*

13. Lines 444-445: It is strange that isoprene has only 5% to the biogenic factor. This indicates that most of the signal in this m/z is probably attributed to other compounds rather than isoprene. Please discuss.

Thera et al.: isoprene and terpenes are known to be biogenic emitted compounds but their biogenic emissions are controlled by different environmental parameters: temperature for terpenes. light and temperature for isoprene (Fuentes et al.. 2000). This implies different diurnal variability of the resulted concentrations. Moreover both compounds show opposite diurnal trends which can also be explained by their different reactivity towards their major oxidants. This is developed in the text in lines 471-474: *[…] Moreover. the diurnal profile of these two compounds show opposite patterns as it can be seen in Figures 5 and 8 which indicates that their biogenic emissions are controlled by different environmental parameters: temperature for terpenes, light and temperature for isoprene (Fuentes et al.. 2000) […].*

The diurnal variability in terpenes in Istanbul with high concentrations at night and early morning and low concentrations during daytime is consistent with the ones already observed in forested or rural areas. This is further developed in the text in lines 480-482: *[…] This type of profile has already been observed at a background site in Cyprus (Debevec et al.. 2017). in a forest of Abies Boriqii-regis in the Agrafa Mountains of north western Greece (Harrison et al.. 2001) and at Castel Porziano near Rome. Italy (Kalabokas et al.. 1997).[…].*

Furthermore, diurnal variation of isoprene and terpenes (see figure below) show that isoprene behave like anthropogenic species which are characterized by a strong increase in concentration at night during period 2 compared to period 1 and 3. This suggests a potential anthropogenic origin for isoprene contrary to terpenes whose variability is poorly affected during period 2.

[Figure]

14. Lines 481. Again. Are you sure it is isoprene?

  Thera et al.: Yes, it is isoprene. M/z 69 in PTRMS has some interferences like furans but the good correlations between m/z 69 and temperature suggests that the contribution of anthropogenic compounds to m/z 69 can be neglected.

15. Lines 486-487: Could you give some examples of "primary biogenic hydrocarbons"?

*Thera et al.:* Some examples of primary biogenic hydrocarbons are: monoterpenes and isoprene. This section was not clear and it has been modified for more clarity in lines 513-517:
*[…]The factor 4 is also characterized by the presence of oxygenated compounds such as isoprene oxidation products like MACR+MVK (54 %) and acetaldehyde (66 %), acetone (57 %), methanol (59 %) and MEK (59 %). These Oxygenated species can have primary sources (both anthropogenic and biogenic) and are also formed secondarily by the oxidation of primary hydrocarbons ( Yáñez-Serrano et al., 2016; Millet et al., 2010; Goldstein and Schade, 2000; Singh, 2004; Schade et al., 2011) […]*

16. Line 497: No. the diurnal profile of the Factor 4 has the opposite behavior according to Fig 8. Please correct the text.

*Thera et al. :* The text is correct but the graph wasn't. The graph has been changed by taking individual contributions of the factors instead of the cumulated contributions of factors which altered some of the prior results. The new graph has been reported below:

[Figure]

**Technical comments:**

*Thera et al.:* All the technical comments has been taken into account.

**Main comment**

The last two sections of the papers (3.3 and 3.4) are rather short but they are potentially important as they compare the results of this study with emission inventories data. Nevertheless, Currently the way the emission ratio is calculated and compared is not convincing at all. The authors say that they can not use the "linear fit regression" method in order to derive emission ratio because there is a poor correlation between targets VOC and CO. They use then the median value of each VOC to CO during all the observation period to estimate an emission ratio (before to compare it to other cities and then to emission inventories). In absence of any correlation between VOC and CO. I do not see how a ratio of median VOC/CO could be used to estimate an emission ratio: : :from what is representative this emission ratio? From all sources for the whole city? Indeed, as the whole dataset is used.,this means that all sources are mixed; and among them traffic contributes only 15%; so how can you compare your ratio to traffic emissions from inventories? As these 2 sections are based on this emission ratio calculation. so either this one is better justified and its representativity (and limitation) is discussed. or these sections have to be removed. We note also that there is no discussion about the fact that VOCs in the inventory stand for "all VOCs" whereas only a limited number of VOC were measured...

Thera et al.: Determining an emission ratio from the slope of a least square linear regression fit is meaningful when VOC and CO correlate which is not the case here as shown by the diurnal profiles and PMF results. The reason is that compounds do not come from the same sources. Indeed, it has been shown by the PMF that traffic was minor. Therefore we propose to calculate the individual ratios between VOCs and CO substracted from their background levels and to derive a statistic representive of this. Deriving a meaningful statistic is not trivial because of the great variability of the individual calculated ratios. At first sight we decided to use the median but the median is not representative of the extreme values that can be found in the ratio especially at night and during period 2. Therefore in this revised version we propose to work on an average emission ratio. From these values we will estimate VOC emissions and an associated standard deviation providing a range of VOC emissions rather than a single value. The contribution of the traffic does not have any impact on the emissions ratio calculation method. We used the following formula to estimate emissions:

$$VOC_{estimated} = ratio\left(\frac{VOC}{CO}\right)_{\substack{all\ observations \\ PMF\ Road\ transport\ factor}} X\ CO_{inventory}$$

For the traffic emission estimation, the VOC/CO was either individual VOC or a sum of VOC present in the family of VOC (like pentanes and xylenes) in PMF road traffic factor. CO inventory is the emission of CO for traffic in the inventories. According to the inventory either ACCMIP or EDGAR, a road transport emission was estimated and compared to the one obtained in the corresponding inventory either by species or by family. For more accurate results we could not use

the sum of the traffic emissions of all VOCs but only individual or a family of species. The emission ratios and evaluation and global inventories section has been improved and the limitations has been discussed as suggested in lines 597-674:

**3.4 Emission ratios of VOC/CO**

*The determination of emission ratios (ER) is a useful constraint to evaluate emission inventories (Warneke et al., 2007; Borbon et al., 2013). The emission ratio is the ratio of a selected VOC with a reference compound that does not undergo photochemical processing mostly CO or acetylene due to their low reactivity at urban scale and as tracers of incomplete combustion (Borbon et al., 2013; Salameh et al., 2017). The linear regression fit method (LRF) is a commonly used method to calculate emission ratios: the ER corresponds to the slope of the scatter plot between a given VOC vs CO or acetylene (Borbon et al., 2013; Salameh et al., 2017). Another method is the photochemical age method (de Gouw, 2005; de Gouw et al., 2018; Warneke et al., 2007; Borbon et al., 2013) which is based on the concentration ratios and the photochemical age. In this study, poor correlation between targets VOC and CO is found ($R^2 \leq 0.16$) as could be deduced from the time series analysis (see section 3.2.2) and the PMF analysis. Indeed, fossil fuel combustion derived activities are not dominating the VOC distribution. As a consequence the LRF method cannot be applied. Here the emission ratio was determined by the mean value of each $\Delta(VOC)$-to-$\Delta(CO)$ concentration ratio over the whole period of measurements. The terms "$\Delta(VOC)$" and "$\Delta(CO)$" correspond to the measured concentrations of VOC and CO subtracted by VOC and CO background concentrations respectively. Given the diurnal and data day-to-day variability of dynamics (see section 3.3.2), one daytime and nighttime CO background values were estimated for each day by extracting the daytime and nighttime minimum concentration values. For CO, the daytime background values range between 213.5 and 367.2 ppb and the nighttime background values range between 211.5 and 406.7 ppb. For VOC, the background values depend on the compound. At night, the background values lie between 1.3 and 3.4 ppb for a long-lived compound like acetone and between 0.2 and 1.1 for a short-lived compound like (m+p)-xylenes. For the following discussion, we will refer to VOC-to-CO ratio instead of "$\Delta(VOC)$-to-$\Delta(CO)$" ratio.*

*Photochemistry can affect the value of emission ratios (Borbon et al., 2013). Comparing daytime to nighttime ratios is one way to evaluate the effect of daytime photochemistry by assuming that chemistry can be neglected at night except for alkenes (de Gouw et al., 2018) and the composition of emissions does not change between day and night. While the ratio between nighttime emission ratios and daytime emission ratios shows a decrease of 37% on average during the day, this decrease is not dependent on the OH kinetic constants of each VOC (Figure S9). This suggest that these differences are rather controlled by the changes in emission composition between day and night. As a consequence, the emission ratios have been determined on the whole dataset.*

*The emission ratios VOC-to-CO in Istanbul are displayed in Table 3 and compared to the ones in other urban areas worldwide. The emission ratios determined in Istanbul are usually higher than the ones of other cities but in the same range of magnitude. C4-C5 alkanes, toluene and oxygenated VOCs show the highest emission ratio values. Most of the values are consistent within a factor of 2 with, at least, one determined in other cities of post-industrialized or developing countries.*

**3.5 Evaluation of global emission inventories**

*In this section, the VOC emissions from anthropogenic sources and road transport source by three references global emission inventories downscaled to Istanbul are evaluated: MACCity (Granier et al.. 2011) for 2014, EDGAR (Crippa et al.. 2018) for 2012, and ACCMIP (Atmospheric Chemistry and Climate Model Intercomparison Project) (Lamarque et al.. 2010) for 2000 (figure 11.a,b and c). Emission data for ACCMIP and MACCity inventories are available in the ECCAD database (http://eccad.aeris-data.fr/), and the one for EDGAR inventory is available in the EDGAR database (http://edgar.jrc.ec.europa.eu/). This evaluation is based on the VOC-to-CO emissions ratios calculated in the previous section (3.4) following Salameh et al. (2016):*

$$VOC_{estimated} = ratio \left(\frac{VOC}{CO}\right)_{\substack{all\ observations \\ PMF\ Road\ transport\ factor}} X\ CO_{inventory} \quad (4)$$

*Where:*

*- VOC estimated is the estimated emission for an individual VOC or a group of VOC in tons/year for all anthropogenic emissions or road transport emissions.*

*- CO inventory is the extracted emission of CO from either ACCMIP (in Tg/year), MACCity (in Tg/year), or EDGAR (in tons/year).*

*- VOC/CO is either the VOC-to-CO ratio calculated in section 3.4 or the VOC-to-CO ratio determined from each VOC contribution in the PMF road transport factor (in $\mu g.m^{-3}$ of VOC/$\mu g.m^{-3}$ of CO).*

*Species in emission inventories are sometimes lumped (grouped) as a function of their reactivity for chemical modeling purpose and species label does not always correspond to a single species. For instance, methanol in Edgar not only corresponds to methanol itself but all alcohols. Moreover, summing some species from observations is sometimes needed to fit with the inventory lumping like alkanes higher than C4 in MACCITY but is limited to the number of the measured species. As a consequence, the comparison is not direct and requires special care (see the following discussion).*

*The annual VOC and CO emissions for EDGAR (0.1 x 0.1 resolution) was determined by summing the emissions of 12 grids over a domain encompassing the sampling site (longitude between 28.9 and 29.1°; latitude between 40.9 and 41.2°). For ACCMIP and MACCITY, the emissions values for the city of Istanbul was taken as available in the ECCAD database.*

*Further information about the emissions inventory will be found in Table S10 of the supplement material.*

*Figure 11 shows the comparison of the estimated emissions of some speciated VOCs derived from observations and PMF for the road transport and the ones from the three global emission inventories downscaled to Istanbul megacity.*

*The total annual VOC anthropogenic emissions by global inventories are usually either within the same range by a factor of two to three for alkanes and aromatics or underestimated by an order of magnitude, especially for oxygenated compounds up to a factor of 58 for acetone by Edgar. These results are consistent with previous evaluations carried out in the Middle East (Salameh et al., 2016) and for northern mid latitude urban areas (Borbon et al., 2013). One exception is methanol in Edgar which is 2.2 times higher than our estimations from observations. This might be due to the inclusion of other alcohols in the methanol label in Edgar as discussed above. One should note that the emissions of CO and VOCs from MACCITY are usually lower than the ones from ACCMIP and EDGAR which can be explained by the different year of reference. The global emissions by inventories were not within the same year: 2000 for ACCMIP, 2014 for MACCITY and 2012 for Edgar. The CO emissions by inventories were compared for the same year. It was found that ACCMIP and MACCITY had the same CO emissions while the emissions in Edgar were two times lower than those from MACCITY and ACCMIP. In 2012, emissions of CO by Edgar was similar to the ones of MACCITY.*

*The evaluation of the road transport emissions (Figure 11.d) is limited to the compounds from the unburned fuel fraction; while there is still an underestimation by the emission inventories except for benzene, the differences are lower than for all anthropogenic emissions. The differences never exceed a factor of 12.1 (pentanes). Again, the differences for pentanes should be seen as a lower limit because of the number of measured pentanes which are limited to n-pentane and isopentane.*

*While these results provide a first detailed evaluation of VOC annual emissions by global emission inventories, they are based on a limited period of observations in September 2014 (2 weeks). Additional VOC observations at different periods of the year including the heating and non-heating period will be very useful to strengthen this first evaluation by taking into account the seasonal variability of emissions. However, they confirm the urgent need in updating global emission inventories by taking into account regional specific emissions.*

**Specific comments:**

-L64: Is the given standard deviation calculated between both calibrations or does it include the 5-ppb control points? How were the calibration coefficients applied to the data? An average value was used or an interpolated one? How was the blank value subtracted? An average value was used or an interpolated one? Please clarify all these points.

Thera et al.: the standard deviation includes the 5-ppb control points and the multi-point values. We clarify all the above mentioned issues in the text in lines 161-165: *[…]The mean calibration factor for all major VOC are derived from the slope of the mixing ratios of the diluted standards with respect to product ion signal normalized to $H_3O+$ and $H_3O+H_2O$. Calibration factors ranged from 2.54 (m/z 137) to 19.0 (m/z 59) normalized counts per seconds per ppbv ($ncps.ppbv^{-1}$). Linearly interpolated normalized background signals are substracted to the normalized signal before applying the calibration factor to determine ambient mixing ratios […]*

-L188: I agree with the author that the variability is highly consistent for aromatics between both techniques. Nevertheless. they claim that the difference in concentrations do not exceed 20%. although the slope for toluene is 22%. In addition, we note that for benzene. there is an underestimation of about 20% of the PTRMS compared to the GC; whereas for toluene. it is the contrary (overestimation by the PTRMS). How do you explain this feature? As ethyl benzene is known to fragment on the mass of benzene. I guess we would rather expect the contrary (i.e. an overestimation of benzene on the PTRMS). Moreover. there seems to exist an even higher difference between the sorbent tube and the GC. As the ratio toluene/benzene is later on used in the paper to comment on source origins. a more careful analysis on the uncertainty associated to this ratio. due to the differences which are pointed out by the intercomparison should be made (as the ratio could be over-estimated).

Thera et al.: The variability between PTRMS and AIRMOVOC is highly consistent (r > 0.85) and the differences in concentrations do not exceed ±22 %. It should be noted that both instruments are calibrated with the NPL and GCU standards respectively. The observed differences takes into account potential differences in calibration factors at least for 10%.

-L190: where is the graph showing the comparison for isoprene?

Thera et al.: the graph showing the comparison for isoprene has been added in the figure S3 of the supplement material and reported below.

[Figure]

-L228: How was calculated the 30-min data? Did it take into account the sampling time of the GC? If not. could it have an impact on the results as there was a high degree of variability of the compounds?

Thera et al.: The 30 minutes was calculated by taking into account the 20 minutes sampling times of the GCFID.

-L244 : With missing values higher than 40% and the use of median values instead of missing data. one can wonder about the meaningfulness of using such compounds? The authors could refer to their sensitivity tests to justify this point.

Thera et al.: As shown in the sensitivity test section, even by removing data with a percentage of missing values above 30%, the PMF results are not changed.

-L251: Even if all details are given in the SM. please give in the main text the values used as input for uncertainties (at least the range)

Thera et al.: The ranges of the input uncertainties has been added in the text in lines 248-249: *[…]The uncertainty of the PTRMS ranges between 5 % (toluene) and 59 % (acetaldehyde) of the concentrations while the uncertainty for the GC-FID ranges between 4 % (2-methyl-pentane) and 17 % (o-xylene) of the concentration[…]*

L265 to 269: I would suggest to move the part in the methodology section -Fig. S4 could be in the main text as it is discussed in details here

Thera et al.: The FLEXPART model description has been moved in the methodology section (section 2.5) in lines 287-293 and the Figure S4 (now Figure 2) has also been moved in the main text.

-L301: "terpens": does it include isoprene?

Thera et al.: No. Terpenes does not include isoprene ($C_5H_{10}$) but $C_{10}C_{16}$ alkenes.

-L307: Give the references associate to the measurements in Paris. London and Beirut

Thera et al.: the references associate to the measurements in paris, London and Beirut have been added in the text in lines 332-333: *[…]Levels of alkanes, some alkenes and aromatics are compared to other European megacities: Paris and London (Borbon et al., 2018) at both urban and traffic site as well as at a suburban site in Beirut (Salameh et al., 2015) during summer[…]*

-L308: one general comment which could be made here is that despite different years and seasons. Istanbul is quite similar to other cities. except for toluene and xylens (and this could be later on reminded when analysing the sources to discuss which of the source(s) would explain these high values in Istanbul)

Thera et al.: this section has been rephrased by taking into account your suggestions in lines 335-339: *[…]Despite differences in absolute levels, the hydrocarbon 335 composition in Istanbul is quite similar to the other cities. Beirut has the highest concentrations of n-butane, isopentane and 2-methyl-pentane. Higher concentrations in toluene, (m+p)-xylenes were in Paris, Beirut and Istanbul. Such similarity would suggest that same sources control the hydrocarbon composition, especially traffic in all cities including Istanbul […]*

-L312: what element suggests the traffic influence? ("This would suggest: : :")

Thera et al.: The similarity of the variability of hydrocarbon composition between different urban areas is the element that can suggest the traffic influence. This feature has been already observed in other cities worldwide. While the absolute levels are different the relative composition is almost the same. This implies that hydrocarbons are controlled by sources of same composition

-L314/Table S6: Why presenting a table of mean concentration which have been measured in different sites (and date/time). It would be more interesting to present a value (or a mean +/-std value) for a given time for each site for some compounds, this would allow a comparison with the main site.

Thera et al.: Table S6 is meant to give an overall view of the concentrations of species measured by tubes and canisters that were not measured by the GCFID nor the PTRMS. Furthermore, lines 346-358 discussed already the relative composition divided into major VOC chemical groups at each sites by sorbent tubes and canisters at different location and date : *[…]The relative composition divided into major VOC chemical groups at each sites by sorbent tubes and canisters is reported on Figure 4a and 4b, respectively. The composition is variable across the megacity for the aliphatic fraction of high and intermediate volatility hydrocarbons (C2-C16). As expected, the composition of the 29/09 12:05 sample at the Besiktas site is like the ones derived from the roadway side measurements highlighting the influence of road transport emissions at the supersite. Interestingly, the VOC composition of the three samples from sorbent tubes at the Besiktas site are different from the ones at the roadway side with a higher proportion of IVOC. The VOC composition of the 26/09 10:31 sample by canister is rather similar to the one from the seashore sample in Galata (29/09 16:12 sample). In the same way the VOC composition of the samples at the supersite derived from tubes are rather like the Besiktas seashore one. For both of them, the proportion of IVOC is significant (from 15 % to 40 % in weight). While light VOC are expected to be of minor importance when considering ship emissions, the higher presence of heavier organics is however expected as observed for alkanes by Xiao et al. (2018) in ship exhaust at berth. The VOC composition comparison would thus suggest not only the impact of road traffic emissions on their composition but also the potential impact of local ship traffic emissions. Finally the composition at Besiktas is not affected by Residential emissions which are enriched in light C2-C3 alkanes (canisters) or aromatics (canisters[…]*

[Figure]

-L331: The section 3.2.2. could be re-arranged. in order to directly introduce the discussion on diurnal variations. In the current version. the overall variability is discussed and then the diurnal variation is discussed but this leads to some confusions (for example, L338 diurnal cycles of NOx

and CO are discussed, although the figures of the diurnal cycles of are not yet properly introduced) and several repetitions (for example, the vegetation type in Istanbul: : :).

Thera et al.: this section has been rearranged in lines 360-441: *[…]The variability of VOC concentrations is driven by several factors: emissions (anthropogenic or biogenic), photochemical reactions (especially with the OH radical during the day and ozone and nitrates at night for alkenes) and the dynamic of the atmosphere (including dilution due to the height of the boundary layer) (Filella and Peñuelas, 2006). The time series of inorganic trace gases (NOx and CO) and some VOC representing the diversity of sources and reactivity are reported in Figure 5. The meteorological periods 1, 2 and 3 described in the previous section 3.1 are also indicated. Because NOx at the super site were only measured from 09/25 to 09/30, data from the air quality station in Besiktas were used (see Figure 1). One should note that the time series of NOx at the supersite and at the Besiktas station are consistent.*

*Time series of NOx and CO show high concentrations but a different pattern regardless of the origin of air masses. While a daily cycle of NOx is depicted, CO does not show any clear pattern. The NO2/NOx ratio fluctuates between 0.34 to 0.93 with an average and median value of 0.53 and 0.55, respectively. These values are very high compared to what is usually found in the literature (Grice et al., 2009; Kousoulidou et al., 2008; Keuken et al., 2012) which are mostly low and below 0.50. However higher values of NO2/NOx ratio can be found in diesel passenger cars (Grice et al., 2009. Vestreng et al., 2009) and vans (Kousoulidou et al., 2008). This ratio would reflect the impact of the combustion of heavy fuels in the megacity. After road transport, cargo shipping is a second highest contributor to NOx levels according to the local/regional inventory (Markakis et al., 2012).*

*Anthropogenic VOC time series (benzene, isopentane and isobutane) exhibit a high frequency variability but usually show higher concentrations during the night especially during period 2. One cause are the very low wind speeds at night especially during period 2 (Figure 3), which would reinforce the accumulation of pollutants. Under marine influence (periods 1 and 3), VOC concentrations are the lowest, especially during period 3, which is characterized by rainy days (September 27th and 28th), high wind speed and colder temperatures (Figure 5). These conditions favor atmospheric dispersion. During transition periods and under continental influence (period 2), VOC concentrations exhibit a strong day-by-day variability with episodic nocturnal peaks especially on September 25th and 26th. While these peaks are not always concomitant between VOC and are not associated with any increase in NOx and CO levels, they occur under south and southwestern wind regimes which are unusual wind regimes according to Figure 1. This points out the potential influence of industrial and port activities other than fossil fuel combustion. For instance, maximum concentrations of butanes occurred during the period of the marine-continental regime shift with well-established southwestern wind regime on 09/22, 09/23 and on 09/26 at the end of the day. Maximum concentrations of pentanes occurred during the night of 09/26 to 09/27 like for aromatics (e.g. benzene) (Figure 5).*

*Except during transition periods, the background levels of measured trace gases are not affected by the origin of air masses. This strongly suggests that the pollutants measured during TRANSEMED-Istanbul were from local and regional sources. Finally, time series would suggest the influence of multiple local and regional sources other than traffic on VOC concentrations, likely industrial and/or port activities, at the supersite.*

*Isoprene and its oxidation products (MACR+MVK) covariate most of the time. They usually show their typical diurnal profiles with higher concentrations during the warmest days and at midday due to biogenic emission processes. Their significant correlation with temperature (R = 0.7) implies the emission from biogenic sources. Around the Besiktas site, 49.5 % of the vegetation is occupied by hardwood and hardwood mix trees while only 6 % is occupied by softwood and hardwood mix trees. While Quercus (isoprene emitter) only occupies 7.7 % of the total vegetation coverage (personal communication from Ministry of Forestry), the presence if isoprene at the supersite is probably due to the surrounding trees.*

*Except during transition periods, the background levels of measured trace gases are not affected by the origin of air masses. The background levels stay constant under continental or marine influence and regardless of the atmospheric lifetime of the species. This strongly suggests that the pollutants measured during TRANSEMED-Istanbul were from local and regional sources. Finally, time series would suggest the influence of multiple local and regional sources other than traffic on VOC concentrations, likely industrial and/or port activities, at the supersite.*

*Taking into consideration time series variability, diurnal variations have been splitted into periods 1 and 3 and period 2 for selected VOC as well as two combustion derived trace gases (NOx and CO). Diurnal profiles of atmospheric concentrations are reported in Figure 6. Local traffic counts for road transport (personal communication from Istanbul Municipality for fall 2014) and ship (https://www.marinetraffic.com/en/ais/details/ports/724/Turkey_port:ISTANBUL) are also reported in Figure S6 in the supplement material. Maritime traffic is mostly for passenger shipping (58.02%) against 16% for cargo shipping. The diurnal profiles of ship and road traffic counts are similar.*

*Generally, concentrations during period 2 are higher than the ones during periods 1 and 3 and show different diurnal patterns for some compounds. The profile of NOx is consistent with the one of traffic counts (Figure S6 of the supplement material). NOx exhibits higher concentrations during the day and lower concentrations at night for both periods with a morning peak (7:30-8:30) and one early evening peak from 17:30 (Figure 6.a). This is typical of traffic emitted compounds with morning and evening rush-hour peaks as observed in many other urban areas like Paris, France in Europe (Baudic et al., 2016) or Beirut, Lebanon in Eastern Mediterranean (Salameh et al., 2016). As already depicted in time series, CO diurnal profile is different from the one of NOx. CO concentrations show higher concentrations in the late evening and lower concentrations during the day. During the day, CO is also characterized by a double peak: one in the morning (8:30) and the other one in the middle of the day (Figure 6.b). Both NOx and CO show quite similar diurnal profile between the three periods even if morning concentrations tend to be higher.*

*VOCs show different profiles from the one of NOx. Under marine influence (periods 1 and 3), primary anthropogenic VOC (ie. benzene, alkanes and other aromatics) almost exhibit a constant profile while they show higher concentration from midnight until 10:00 AM under continental influence (period 2), For instance, benzene (Figure 6.d) and isopentane (Figure 6.f) nighttime concentrations increase by four-fold compared to the levels under marine influence. In the middle of the day, the concentration levels are the same as during periods 1 and 3. The profiles of primary anthropogenic VOCs point out the complex interaction between local and regional emissions and dynamics. Period 2 points out the influence of VOC emissions other than traffic and combustion processes (no effect on NOx and CO) at night. While the influence of traffic emissions on CO and VOC cannot be excluded; it seems that their emission level is not high enough to counteract the dispersion effect during the day unlike NOx. This will be further investigated in the PMF analysis.*

*Isoprene concentrations increase immediately at sunrise and decrease at sunset during period 1 and 3 (marine influence) which indicates its well-known biogenic origin (Figure 6.g) which is light and temperature dependent. Isoprene and MACR+MVK's concentrations increase at night during period 2 like other alkanes and aromatics, suggesting their potential anthropogenic influence.*

*Provided some interferences like furans could contribute to isoprene signal by PTRMS measurements (Yuan et al., 2017), this would suggest an anthropogenic origin for isoprene. While the signals of m/z 71 are commonly attributed to the sum of MVK and MACR which are both oxidation products of isoprene under high-NO conditions, more recent GC-PTR-MS studies identified some potential interferences for MVK and MACR measurements, including crotonaldehyde in biomass-burning emissions, C5 alkenes, and C5 or higher alkanes in urban regions (Yuan et al., 2018). Such interferences cannot be ruled out here. During periods 1 and 3, MACR+MVK concentrations follow the same general pattern as of isoprene's.*

*With relatively long atmospheric lifetime, (≈ 68 days), acetone's concentration is quite constant throughout the day within period 1, a peak in the middle of the day and lower concentrations during the night for period 2 (Figure 6.c). The peak in the middle suggests the presence of a secondary origin. Acetone can have both primary and secondary source (Goldstein and Schade, 2000; Macdonald and Fall, 1993). Methanol and MEK have the same general pattern as for acetone during both periods without the peak in the middle of the day suggesting that they might have the same emission source[…].*

-L338 and the corresponding paragraph: The discussion of this section is not clear and might be improved, once the discussion includes as well the diurnal cycles (see previous comment). In addition the discussion focusses mainly on local meteorological conditions (wind. dispersion..) but

no discussion is made on the possible influence of long-range transport. If not discussed at all, why studying Flexpart back-trajectories over such long periods?

Thera et al.: The discussion of this section has been improved more clarity in lines 360-441. The objective of studying Flexpart was to see air mass trajectory. The Time series of our species did not enabled us to the see long range transport since we couldn't distinguish long or local range transport.

-L338: At midday it is not a maximum. In addition, why a midday concentrations max is expected from traffic-related compounds, Usually a morning and an evening peak are observed

Thera et al.: this section has been corrected and rearranged in lines 409-413: *The profile of NOx is consistent with the one of traffic counts (Figure S6 of the supplement material). NOx exhibits higher concentrations during the day and lower concentrations at night for both periods with a morning peak 410 (7:30-8:30) and one early evening peak from 17:30 (Figure 6.a). This is typical of traffic emitted compounds with morning and evening rush-hour peaks as observed in many other urban areas like Paris, France in Europe (Baudic et al., 2016) or Beirut, Lebanon in Eastern Mediterranean (Salameh et al., 2016).*

-L341: Isoprene and its oxidation products co-variate most of the time. This is not true for period 2. Be more precise in your analysis and description.

Thera et al.: Precision has been made in this section in lines 391-392: *[…]Isoprene and its oxidation products (MACR+MVK) covariate most of the time. They usually show their typical diurnal profiles with higher concentrations during the warmest days and at midday due to biogenic emission processes[…].*

-L384 and Figure 5: there is a large peak of benzene. isopentane. isobutene. m71 during the night of event 2. How do you interpret it? Is it due to a single event or it was observed several times? It could be useful to show toluene on this figure (directly near to benzene)

Thera et al.: during period 2 large peaks are observed during several nights for many VOCs. It is probably due the wind regimes (low wind speed that will favor the accumulation of pollutants). Moreover they occur under south and southwestern wind regimes which correspond to unusual wind regimes according to Figure 1This points out the potential influence of industrial and portactivities other than fossil fuel combustion as detailed in the Time Series section in lines 375-377: *[…]Anthropogenic VOC time series show highest concentrations during the night especially during period 2. One cause are the very low wind speeds at night especially during period 2 (Figure 3), which would reinforce the accumulation of pollutants […]*
We did not show toluene directly near benzene because it will be difficult to show all the compounds and we also have a specific PMF factor for toluene in section 3.3 where its diurnal profile and time series are discussed.

-L406 to L418: I would suggest to move this part in the methodology section

Thera et al.: This section has been moved in the methodology section in lines 260-271: *[…]PMF reference run has been performed by removing the period during which there were no GC-FID data (night from 09/24 to 09/25). In addition, these data set have been chosen as PMF reference run because of the higher correlation between observed and reconstructed data by the PMF model (see also section 3.2). A good correlation (R2 = 0.97) between total reconstructed VOC and measured VOC was obtained. For most compounds the variability is well reproduced with an R2 usually higher than 0.70. Poorer correlation was found for alkenes (1-pentene (R2 = 0.55), 1,3-butadiene (R2 = 0.22) and isoprene (R2 = 0.57) as well as for n-hexane (R2 = 0.09), MEK (R2 = 0.41) and acetaldehyde (R2 = 0.32). Moreover, the R2 between the five factors does not exceed 0.28 and is usually less than 0.05 indicating the statistical independence of the five factors. The R2 of the contribution of the five factors between each other has been calculated, it was found that the value of R2 does not exceed 0.28. There is therefore no significant correlation between the factors which means that the factors are independent.*

*The PMF output uncertainties were estimated by three models: the DISP model (base model displacement error estimation), the BS model (base model bootstrap error estimation) and the DISP+ BS model. Further information for the estimation of model prediction uncertainties can be found in Norris et al. (2014) and Paatero et al. (2014). The DISP results of the PMF run show that the 5-factor solution is stable and sufficiently robust to be used because no swaps occurred. All the factors were well reproduced through the BS technique at 100 % for factor 1, 96 % for factor 2, 100 % for factor 3, 99 % for the factor 4 and 100 % for factor 5; there were not any unmapped run. The DISP+BS model shows that the solution is well constrained and stable[…].*

-L422: why naming a source after a compound and not only "solvent use"?

Thera et al.: We named the source after a compound and not only solvent use because even though toluene is the main compound in this factor (57 %), it also contribute up to 29 % to the road transport factor.

-L422: The recent study about VOCs from petrochemical sources in urban areas (Mac Donald et al;. Science. 2018) must be referenced somewhere when discussing about solvent use

Thera et al.: Mac Donald et al., (2018) has been referenced in the last section while discussing about the PMF results as a whole in lines 587-592: *[…]these differences in contributions with this study could be due to the differences in input data. Thus, PMF results depends strongly on input data. Furthermore, it was shown in McDonald et al. (2018) that source apportionment studies largely underestimated the influence of Volatile Chemical Species (including organic solvents, personal care products, adhesives …) as source of urban VOC. This underestimation could be explained by the fact that VOC are not measured in all their diversity in source apportionment studies in contrast with what was done in McDonald et al., (2018) […].*

-L432: The sentence "low T/B ratio indicates the influence of traffic emissions on measured VOCs: : :.." could be mis-leading and should be checked /re-formulated (see for example Gaeggeler et al.. 2008 which says the opposite: "Another indicator for traffic emissions is a low benzene/toluene ratio (Stemmler et al.. 2002)". In addition, the uncertainty of the T/B ratio should be reminded here (see comment L188). Therefore, this section should be either removed or discussed more thoroughly.

Thera et al.: Low B/T (0.38) in Stemmler et al. (2002) correspond to our low T/B ([2-3]). Indeed as you said the sentence could be misleading. The section has been rephrased with more clarification and uncertainties of T/B prior to the GCFID and PTRMS have been added in lines 458-469: *Toluene/benzene ratio (T/B) is used as an indicator of non-traffic source influence (Elbir et al., 2007; Lee et al., 2002; Yurdakul et al.,2013). T/B ratio ≤ 2-3 indicates the influence of traffic emissions on measured VOC concentrations (Gelencsér et al.,1997; Heeb et al., 2000; Muezzinoglu et al., 2001; Brocco et al., 1997) whereas T/B ratios ≥ 2-3 suggests the influence of other sources than traffic (such as solvent evaporation or industrial sources). The T/B ratio for this study is between 0.4 (with only 4 points below 2) and 48.6 (Only 1 point above 29). Only 5.8 % of the ratios were between 2 and 3, 48 % were between 3 and 6 while 45 % were above 6 with 34 % between 6 and 10. This strongly suggests the influence of sources of toluene other than traffic. High value of T/B ratio is mostly found at industrial sites (Pekey and Hande, 2011). The median and mean value of T/B in this experiment are respectively 5.6 and 6.7 which can also indicate gasoline related emissions (Batterman et al., 2006). However the absence of other unburned fuel compounds like pentanes excludes this source. These ratios were calculated with toluene and benzene measured by the PTRMS since the PMF run was done by those data. By looking at the T/B ratio measured by the GCFID, we found approximatively the same conclusion: Only 1 % of the ratios were between 2 and 3, 47 % were between 3 and 6 while 51 % were above 6 with 38 % between 6 and 10. This factor represents 14.2 % of the total contribution.*

-L477: could this factor represents the "regional background"? If so. the discussion could be shortened. as there is no specific source associated and therefore no need to detail all biogenic/anthropogenic. primary/secondary source. That would avoid some vague statement. For example. L486 "these species are formed by the oxidation of primary biogenic hydrocarbons. However these oxygenated can have also primary both anthropogenic and biogenic sources". And the mention of 1.3-butadiene and 1-pentene being emitted by plants is not so convincing in such a highly populated city.

Thera et al.: Since this factor has a large contribution of isoprene which is reactive (lifetime less than 2 hours), it cannot be assigned to a "regional background" factor. To name a factor after regional background there must only be species with long lifetime so, which low reactivity; which is not the case in our study : we have a mixed of species of low and high reactivity see Baudic et al. (2016) and of different primary and secondary. Moreover one would expect this background to increase during period 2 (continental influence) which is not the case except during the first transition period.

-L480: the sensitivity study should be mentioned here (otherwise the 70% missing value would lead to the comment that this compound should not be taken into account).
Thera et al.:  The sentitivity study has been moved to the methodology section in section 2.4.5.

L517: it is difficult to see on the figure that a strong increase in minimum concentrations is observed during period 2
Thera et al.: Strong has been removed and replaced by "an increase in minimum concentration" and the graph has been changed from cumulative contribution of factors time series

by simple contributions of factors. The graph is reported below and has also been changed in the article.

[Figure]

-L549: This sentence is too vague; how has it been analyzed? Either remove or give a bit more information on this point.

Thera et al.: More information has been added in this section in lines581-583: *[...]As it was discussed in Yuan et al., (2012), the effect of photochemistry on factors composition had been analyzed by looking at the scatterplots of the contribution of the PMF factors to each VOC as a function of its OH rate constant (k_OH). Nevertheless, no clear evidence from photochemistry was founded on the Istanbul PMF factor's contributions. [...]*

-L551: This section on sensitivity tests is important and is convincing to show that the most appropriate run has been selected. As these results are needed before. I'm wondering if it would not be more appropriate to move it at the beginning of the PMF results section (or even in the methodology part). The second part of the section (starting from L560) does not really belong to a section called "sensitivity tests" and it is not clear what it brings to the discussion. Therefore. It is suggested either to remove it or to discuss it in more details (probably in another section then).

Thera et al.: The sensitivity tests section have been moved in the methodology section. The second part of the second has been deleted as you suggested.

-L551: Before to start a new section, it would be useful to have a section which comments the PMF results as a whole (for example. the contribution of the different sources compared to the other cities where levels and variability were compared: : :)

Thera et al.: A comparison of PMF factors between our study and some cities where levels and variabilities were compared has been made in lines 584-596: *[...] This study show that PMF was able to extract easily some factors (like biogenic terpenes) than others (like diurnal regional factors). These results are consistent with other Turkish cities where other source than traffic (mostly industrial source) drive the VOC emissions (Yurdakul et al., 2013; Pekey and Hande, 2011; Civan et al., 2015; Dumanoglu et al., 2014). However, in the EMB, traffic related emissions are the most dominant source and accounted for 51 and 74 % in winter and summer respectively in Beirut, Lebanon (Salameh et al., 2016). Kaltsonoudis et al. (2016) also found that traffic and biogenic emissions were the dominant source of VOC during summer in Patras and Athens. In Paris, Baudic et al. (2016) found that 25 % of the total VOC contributions were related to traffic, 15 % to biogenic factor, 20 % to solvent use against 14.2 % and 23% to natural gas and background factor that the PMF has not able to dissociate. These differences in contributions with this study could be due to the differences in input data. Thus, PMF results depends strongly on input data. Furthermore, it was shown in McDonald et al. (2018) that source apportionment studies largely underestimated the influence of Volatile Chemical Species (including organic solvents, personal care products, adhesives ...) as source of urban VOC. This underestimation could be explained by the fact that VOC are not measured in all their diversity in source apportionment studies in contrast with what was done in McDonald et al. (2018). [...]*

**Technical comments:**

Thera et al. : All the technical comments has been taken into account.